# Distributional Adversarial Attacks and Training in Deep Hedging

**Guangyi He**
Department of Mathematics
Imperial College London
g.he23@imperial.ac.uk

**Tobias Sutter**
Department of Economics
University of St. Gallen
tobias.sutter@unisg.ch

**Lukas Gonon**
School of Computer Science
University of St. Gallen
lukas.gonon@unisg.ch
Department of Mathematics
Imperial College London
l.gonon@imperial.ac.uk

## Abstract

In this paper, we study the robustness of classical deep hedging strategies under distributional shifts by leveraging the concept of adversarial attacks. We first demonstrate that standard deep hedging models are highly vulnerable to small perturbations in the input distribution, resulting in significant performance degradation. Motivated by this, we propose an adversarial training framework tailored to increase the robustness of deep hedging strategies. Our approach extends pointwise adversarial attacks to the distributional setting and introduces a computationally tractable reformulation of the adversarial optimization problem over a Wasserstein ball. This enables the efficient training of hedging strategies that are resilient to distributional perturbations. Through extensive numerical experiments, we show that adversarially trained deep hedging strategies consistently outperform their classical counterparts in terms of out-of-sample performance and resilience to model misspecification. Additional results indicate that the robust strategies maintain reliable performance on real market data and remain effective during periods of market change. Our findings establish a practical and effective framework for robust deep hedging under realistic market uncertainties.[1]

## 1 Introduction

In the past decades, options and derivatives trading has seen enormous growth, with billions of contracts traded every year [1]. Hedging of financial derivatives refers to taking financial positions in other linked assets to mitigate the risk associated with the derivative. Thereby, hedging is a fundamentally important task in the financial services industry. Deep Hedging [2] introduces a model-free, data-driven framework that employs deep learning to optimize hedging strategies directly from simulated market scenarios. The core idea is to parameterize hedging decisions via neural networks. Optimal hedging is achieved by training the neural networks to minimize a risk measure of the hedged portfolio's profit and loss (P&L). While this approach has been widely adopted in industry, the performance of learned hedging strategies critically depends on the quality of the training data. For instance, the data-generating distribution during training might differ slightly from the test distribution under which the deep hedging strategy is ultimately deployed and evaluated. This mismatch, known as *model misspecification*, can result in suboptimal or even misleading decisions, a phenomenon that is well documented in various classes of data-driven decision-making problems [3, 4]. This

---

[1]The code is available on https://github.com/Guangyi-Mira/Distributional-Adversarial-Attacks-and-Training-in-Deep-Hedging

39th Conference on Neural Information Processing Systems (NeurIPS 2025).

highlights the fundamental challenge of modeling uncertainty in finance and the importance of designing strategies that remain robust to small changes in the underlying distribution.

To mitigate model uncertainty in decision-making problems, a large variety of different approaches exist. Parametric methods quantify uncertainty on model parameters, allowing strategies to account for potential estimation errors, like in [5, 6]. Moreover, advances in machine learning enable the use of generative models with high-dimensional parameter spaces to simulate complex dynamics [7, 8, 9]. In the field of deep hedging, recent works consider robustness by perturbing the terminal distribution [10], randomized model parameters [11] or penalizing deviations from a reference distribution [12].

On the other hand, model uncertainty can be addressed within the framework of *distributionally robust optimization* (DRO) [13]. DRO seeks to make optimal decisions under the worst-case scenario over a specified set of probability distributions, known as the *ambiguity set*. This problem can be formulated as a two-player game

$$\min_{\theta \in \Theta} \ \max_{\eta \in B_\delta(\mu)} \ \mathbb{E}_{x \sim \eta}\big[l(\theta; x)\big], \tag{1.1}$$

where $\theta$ denotes the decision variable optimized over the parameter space $\Theta$, representing the first player. The adversarial player selects a distribution $\eta$ from an ambiguity set $B_\delta(\mu)$ of plausible distributions centered around a nominal distribution $\mu$. The objective is described by the loss function $l(\theta; x)$, parameterized by $\theta$ and evaluated at an input $x \in \mathbb{R}^d$, which is modeled as a random variable. The parameter $\delta > 0$ controls the size of the ambiguity set, with $B_0(\mu) = \{\mu\}$. Thus, when $\delta = 0$, the DRO problem (1.1) reduces to the nominal stochastic optimization problem

$$\min_{\theta \in \Theta} \ \mathbb{E}_{x \sim \mu}\big[l(\theta; x)\big]. \tag{1.2}$$

Various functions to encode the difference between probability distributions in $B_\delta(\mu)$ are popular, such as the $\phi$-divergence [14], matching moments [15], and the total variation distance [16]. Among these, the Wasserstein distance is particularly popular and powerful, see the appendix for the detailed definition and see [17, 18, 13] for detailed treatments of Wasserstein-based DRO problems. These existing computational tractability results in Wasserstein DRO rely on structural assumptions about the underlying loss function, such as convexity, which do not hold in the deep hedging setting considered in this work.

The concept of robustness is important across machine learning applications. In image classification tasks, [19] first highlighted that neural networks are vulnerable to adversarial attacks—small, imperceptible changes to input images that cause misclassification. This discovery led to a growing body of research on adversarial attacks and training techniques to improve network robustness against them ([20, 21, 22]). However, most studies have focused on pointwise adversarial perturbations, where attacks modify individual data points. A broader perspective considers distributional attacks, where an adversary perturbs the entire data-generating distribution rather than specific samples ([23, 24, 25, 18, 13, 26, 27]). Specifically, the distributional version of adversarial attacks can be applied to search for the worst-case scenario in the DRO problem.

In this paper, we study the robustness properties of classical deep hedging strategies under distribution shifts by utilizing the concept of adversarial attacks. The main contributions of this paper can be summarized as follows:

- We propose the first framework that unifies distributional adversarial training with deep hedging. Building on tractable reformulations from Wasserstein DRO [26] and sensitivity analysis results [23], our approach derives computationally efficient adversarial attacks within a Wasserstein ball and integrates them into an adversarial training procedure. This framework reveals the vulnerability of classical deep hedging to even small distributional shifts and provides a tractable method to enhance robustness.

- Prior work [13, 17, 18] establishes that data-driven Wasserstein DRO solutions guarantee out-of-sample performance under certain theoretical assumptions. However, in complex dynamic settings such as deep hedging, these assumptions do not generally hold. Through adversarial training with limited data (5,000–100,000 samples), we empirically demonstrate that adversarially trained strategies outperform classical deep hedging methods in terms of out-of-sample and out-of-distribution performance.

- We validate our framework on real market data, demonstrating that adversarially trained strategies remain effective and robust during periods of market stress and distributional shifts, and compare our results against the robust deep hedging approach of [11].

**Related work.** Deep hedging was introduced in [2, 28]. Since its introduction, numerous works have studied deep hedging and related machine learning-based hedging methods from different perspectives. Robustness of deep hedging methods has been studied in [10, 11, 12]. Further directions include tackling complex pricing problems [29, 30, 31], targeting improved efficiency [32, 33] or general initial portfolios [34], alternative reinforcement learning frameworks [35, 36, 37, 38], empirical approaches [39, 40], or implementations on quantum hardware [41]. We refer to the surveys [42, 43] for a comprehensive overview.

Parallel to this, distributionally robust optimization (DRO) has emerged as a framework for decision-making under model uncertainty, with Wasserstein-based methods offering both theoretical guarantees and computational tractability [13, 17, 18]. In machine learning, adversarial training provides another robustness perspective, where worst-case perturbations improve generalization [19, 20, 21, 22]. Recent works extend pointwise adversarial attacks to the distributional setting [23, 24, 25, 26, 27], thereby connecting adversarial training methods with DRO formulations.

## 2 Deep Hedging Framework

**Basic setup.** In Deep Hedging [2], we consider a discrete-time market with trading dates $\{0, 1, \ldots, T\}$ and a set of $r$ tradable assets. Let $\mathbf{S} = (S_1, \ldots, S_{T+1}) \in \mathbb{R}^{r \times (T+1)}$ denote the mid-price trajectories of the assets. The price process $\mathbf{S}$ is adapted to a filtration $(\mathcal{F}_t)_{t=0}^T$ generated by an information process $\mathbf{I} = (I_1, \ldots, I_{T+1})$, where each $I_t \in \mathbb{R}^d$ captures market-relevant features at time $t$ such as asset prices, volatility, risk limits, or trading signals. We denote the price trajectories $\mathbf{S}(\mathbf{I})$ as a function of $\mathbf{I}$. The trajectory $\mathbf{I}$ is sampled from a known distribution $\mu$, which may correspond to a stochastic model or a uniform distribution over simulated or historical scenarios. A trading strategy $\delta = (\delta_1, \ldots, \delta_{T-1}) \in \mathbb{R}^{r \times T}$ represents the portfolio holdings in the $r$ assets. Each position $\delta_t$ is determined by a neural network that is parameterized by $\theta = (\theta_1, ..., \theta_T)$ and takes the history of available information up to time $t$ as input,

$$\delta_t = f_{\theta_t}(I_1, \ldots, I_t), \tag{2.1}$$

where $f_\theta$ is a function parameterized by network weights $\theta_t \in \mathbb{R}^n$. Therefore, the whole trading strategy $\delta$ depends on the the network parameter $\theta \in \Theta$ and information process $\mathbf{I}$, denoted $\delta(\theta, \mathbf{I})$.

We consider hedging a contingent claim with payoff $P(S_T)$ at maturity, where $P: \mathbb{R}^r \to \mathbb{R}$ is a given payoff function. The profit and loss (PnL) of the hedging position is defined as

$$\mathrm{PnL}(\theta, \mathbf{I}) = p_0 + \sum_{t=1}^T \delta_t^\top(\theta, \mathbf{I})(S_{t+1}(\mathbf{I}) - S_t(\mathbf{I})) - P(S_T(\mathbf{I})) \tag{2.2}$$

for a given price $p_0 \geq 0$. The objective is to optimize the hedging outcome $\mathrm{PnL}(\theta, \mathbf{I})$ under a convex risk measure $\rho$, such as the entropic risk measure or Conditional Value at Risk (CVaR) [44], i.e., solve

$$\min_{\theta \in \Theta} \rho\left[\mathrm{PnL}(\theta, \mathbf{I})\right]. \tag{2.3}$$

**Optimized Certainty Equivalents.** We focus on convex risk measures that admit an optimized certainty equivalent (OCE) form. Specifically, consider convex risk measures that can be expressed as

$$\rho(Z) = \inf_{\omega \in \mathbb{R}} \left\{\omega + \mathbb{E}\left[\ell(-Z - \omega)\right]\right\}, \tag{2.4}$$

for random variables $Z$ and a fixed loss function $l: \mathbb{R} \to \mathbb{R}$. OCE risk measures form an important class of convex risk measures [45], naturally suited for deep hedging [2]. During training, we implement an OCE risk measure by treating $\omega$ as a trainable parameter. That is, we consider an augmented parameter $\tilde{\theta} := (\theta, \omega)$ and define the deep hedging loss as

$$l_{\mathrm{DH}}(\tilde{\theta}, \mathbf{I}) = \omega + \ell(-\mathrm{PnL}(\theta, \mathbf{I}) - \omega). \tag{2.5}$$

We can equivalently express the deep hedging problem (2.3) as a standard expected loss minimization

$$\min_{\tilde{\theta}} \mathbb{E}_{\mathbf{I} \sim \mu}[l_{\mathrm{DH}}(\tilde{\theta}, \mathbf{I})]. \tag{2.6}$$

During the evaluation, the optimal $\omega$ can be obtained by directly solving (2.4).

**DRO Formulation of Deep Hedging.** The reformulated deep hedging problem (2.6) now has the common structure of (1.2), which induces a corresponding DRO formulation

$$\min_{\tilde{\theta}} \max_{\eta \in B_\delta(\mu)} \mathbb{E}_{\mathbf{I} \sim \eta}[l_{\mathrm{DH}}(\tilde{\theta}, \mathbf{I})]. \tag{2.7}$$

The OCE framework naturally encompasses common convex risk measures such as entropic risk and Conditional Value-at-Risk (CVaR), whose specific implementations will be demonstrated in subsequent examples. Crucially, the OCE reformulation aligns with standard DRO formulations, enabling direct utilization of established convergence guarantees and computational methods.

**Examples.** We consider a Black-Scholes model and General Affine Diffusion model with entropic risk and a Heston model with CVaR. Detailed formulations of these models and the associated loss functions are provided in the appendix.

## 3 Adversarial Attacks

**Pointwise adversarial attacks.** Pointwise adversarial attacks aim to perturb the input within a $\delta$-ball centered at the input to maximize the corresponding loss, that is,

$$\max_{\hat{x} \in \mathbb{R}^d} l(\theta; \hat{x}) \quad \text{subject to} \quad d(\hat{x}, x) \leq \delta. \tag{3.1}$$

Two cornerstone algorithms in this domain are the Fast Gradient Sign Method (FGSM) [20] and Projected Gradient Descent (PGD) [21]. For input $x$ and $L^\infty$ distance $d(\cdot, \cdot)$, the FGSM perturbation is computed as

$$\hat{x} = x + \delta \cdot \mathrm{sign}(\nabla_x l(\theta; x)), \tag{3.2}$$

where $\delta$ is the perturbation magnitude, $\nabla_x l(\theta; x)$ is the gradient of the loss function $l$ with respect to the input $x$, and $\mathrm{sign}(\cdot)$ denotes the element-wise sign function. FGSM directly perturbs the input data in the direction of the gradient to the boundary of the $\delta$-ball around the original input $x$, which is computationally efficient. For a deeper exploration of the worst-case perturbation, PGD is an iterative extension of FGSM that applies multiple small perturbations to the input. At each iteration, the input is updated as

$$\hat{x}^{(k+1)} = \mathrm{Proj}_{d(\cdot, x) \leq \delta}\left(\hat{x}^{(k)} + \beta \cdot \mathrm{sign}(\nabla_x l(\theta; \hat{x}^{(k)}))\right), \tag{3.3}$$

where $\beta$ is the step size, and $\mathrm{Proj}_{d(\cdot, x) \leq \delta}$ projects the perturbed input back into the $\delta$-ball around $x$ to ensure the perturbation remains bounded.

**Distributional adversarial attacks.** Distributional adversarial attacks aim to find the worst-case data distribution perturbation within an ambiguity set $B_\delta(\mu)$ that maximizes the expected loss of the model. This is formalized as

$$V_\theta(\delta) = \max_{\eta \in B_\delta(\mu)} \mathbb{E}_{x \sim \eta}[l(\theta; x)], \tag{3.4}$$

where an optimal perturbed distribution $\eta^*$ achieves this maximum. Solving (3.4) requires to optimize over an infinite-dimensional Wasserstein ball, making direct approaches intractable. Therefore, we provide a reformulation that approximates (3.4) based on the sensitivity results on DRO in Wasserstein balls that are inspired by [46]. The proposed reformulation relies on the following two assumptions.

**Assumption 3.1.** *The distance $d$ on $\mathbb{R}^d$ is induced by a norm $\|\cdot\|$, with corresponding dual norm $\|\cdot\|_*$ defined as*

$$\|y\|_* = \sup_{\|x\| \leq 1} \langle y, x \rangle. \tag{3.5}$$

*Furthermore, there exists a function $h : \mathbb{R}^d \to \mathbb{R}^d$ such that*

$$\|x\|_* = \langle h(x), x \rangle \quad \text{for all } x \in \mathbb{R}^d. \tag{3.6}$$

**Assumption 3.2.** *For each $\theta$, the map $x \mapsto l(\theta; x)$ is Lipschitz continuous.*

Assumption 3.1 is satisfied, e.g., for $d$ chosen as the distance induced by the $\|\cdot\|_p$ norm. In this case, the dual norm is given by $\|\cdot\|_q$, where $q$ is the Hölder conjugate satisfying $1/p + 1/q = 1$. The corresponding function $h : \mathbb{R}^d \to \mathbb{R}^d$ is explicitly given (with operations applied component-wise) as

$$h(x) = \mathrm{sign}(x) |x|^{q-1}. \tag{3.7}$$

Additionally, we focus on the data-driven framework, where the input distribution $\mu$ is the empirical distribution constructed from the training set $\{X_1, \ldots, X_N\}$, i.e., $\mu = \frac{1}{N} \sum_{n=1}^{N} \delta_{X_n}$.

Analogously to [23, Theorem 4.1] in the setting of a classification problem, we use [46, Theorem 2.1] and its proof to establish the following theorem. The details are provided in the appendix.

**Theorem 3.3.** *Under Assumption 3.1 and Assumption 3.2, $V_\theta(\delta)$ can be approximated by*

$$V_\theta(\delta) = \mathbb{E}_{\eta_\delta}[l(\theta, x)] + o(\delta) \quad as \ \delta \downarrow 0, \tag{3.8}$$

*where $\eta_\delta = \frac{1}{N} \sum_{n=1}^{N} \delta_{\hat{X}_n}$. For $n = 1, \ldots, N$, each $\hat{X}_n$ is the perturbation of the original sample*

$$\hat{X}_n = X_n + \delta \cdot h(\nabla_x l(\theta; X_n)) \|\nabla_x l(\theta; X_n)\|_*^{q-1} \Upsilon^{1-q}, \tag{3.9}$$

$$\Upsilon = (\frac{1}{N} \sum_{n=1}^{N} \|\nabla_x l(\theta; X_n)\|_*^q)^{1/q}. \tag{3.10}$$

Equation (3.9) provides an approximate worst-case perturbation by leveraging the sensitivity expansion of the Wasserstein DRO problem, which directly yields a computationally tractable solution that approximates (3.4). However, this approach may not fully explore the adversarial landscape. Therefore, we instead constrain optimization to a subset of the ambiguity set which contains $\eta_\delta$ and has a simpler structure. The set $\hat{B}_\delta(\mu_N)$ is defined as

$$\hat{B}_\delta(\mu_N) = \left\{ \hat{\mu}_N = \frac{1}{N} \sum_{n=1}^{N} \delta_{\hat{X}_n} : \hat{X}_i \in \mathbb{R}^d, (\frac{1}{N} \sum_{i=1}^{N} d(X_i, \hat{X}_i)^p)^{1/p} \leq \delta \right\}. \tag{3.11}$$

**Lemma 3.4.** *Consider the empirical optimization problem*

$$V_\theta^e(\delta) = max_{\eta \in \hat{B}_\delta(\mu)} \mathbb{E}_{x \sim \eta}[l(\theta; x)], \tag{3.12}$$

*obtained by replacing the ambiguity set $B_\delta(\mu)$ in (3.4) by $\hat{B}_\delta(\mu)$ as defined in (3.11). Under Assumptions 3.1 and 3.2, we have*

$$V_\theta(\delta) = V_\theta^e(\delta) + o(\delta) \quad as \ \delta \downarrow 0. \tag{3.13}$$

The set $\hat{B}_\delta(\mu_N)$ has been previously studied in the literature [26, 27] with focus on the setting where $N$ grows. Our paper provides a sensitivity analysis as $\delta \downarrow 0$. Finally, we write (3.12) into a constrained problem perturbing the dataset directly

$$V_\theta^e(\delta) = max_{\hat{X}_1, \ldots, \hat{X}_N} \frac{1}{N} \sum_{n=1}^{N} l(\theta; X_n) \quad \text{subject to} \quad (\frac{1}{N} \sum_{i=1}^{N} d(X_i, \hat{X}_i)^p)^{1/p} \leq \delta. \tag{3.14}$$

It is worth noticing that, as $p \to \infty$, the constraint in (3.14) becomes an $L^\infty$-constraint, i.e., $max_n(d(X_n, \hat{X}_n)) \leq \delta$, which coincides with the pointwise adversarial attack setting.

## 4 Algorithms for Distributional Adversarial Attacks in Deep Hedging

In this section, we develop a numerical algorithm to solve the distributional adversarial attack problem (3.4) via its tractable approximation (3.14) in the deep hedging framework. Drawing inspiration from adversarial training methods in machine learning, we adapt the Fast Gradient Sign Method (FGSM) [20] and Projected Gradient Descent (PGD) [21] to the distributional setting. For each algorithm, we first propose a one-step FGSM-like perturbation to generate adversarial distributions efficiently. We then extend this to a multi-step PGD-like algorithm, which iteratively refines the perturbation with smaller steps while constraining the perturbation through a projection step.

We first focus on the Black-Scholes model, where the input information is exactly the single price trajectory $\mathbf{S}$, as detailed in the appendix. We will then explain how to extend the algorithm to models with more than one trajectory, including the Heston model, where the input is a pair of price and variance trajectories $\mathbf{S}$ and $\mathbf{v}$. For the readers' convenience, we focus on these two cases here and present algorithms for a general setting in the supplementary material.

In deep hedging, with the loss function $l_{DH}(\theta; \mathbf{S})$ as defined in (2.5), problem (3.14) becomes

$$max_{\hat{\mathbf{S}}_1, \ldots, \hat{\mathbf{S}}_N} \frac{1}{N} \sum_{n=1}^{N} l_{DH}(\theta; \hat{\mathbf{S}}_n) \quad \text{subject to} \quad (\frac{1}{N} \sum_{n=1}^{N} d(\mathbf{S}_n, \hat{\mathbf{S}}_n)^p)^{1/p} \leq \delta. \tag{4.1}$$

We define the distance in the input space $\mathbb{R}^{T+1}$ as the $L^\infty$ distance, i.e.,

$$d(\mathbf{S}_n, \hat{\mathbf{S}}_n) = \|\mathbf{S}_n - \hat{\mathbf{S}}_n\|_\infty = \max_{t=0,\ldots,T} |S_{n,t} - \hat{S}_{n,t}|. \tag{4.2}$$

The infinity norm has the $L^1$-norm as its dual norm, i.e., $\|g\|_* = \sum_{t=0}^T |g_t|$, and $h(x) = \text{sign}(x)$, applied component-wise, satisfies $\langle h(x), x \rangle = \|x\|_*$, which implies Assumption 3.1.

**Wasserstein Projection Gradient Decent (WPGD).** Following the above setup, equation (3.9) provides an asymptotic approximation of the worst-case distributional shift, leading to a one-step perturbation of each path like FGSM

$$\mathbf{S}_n \mapsto \mathbf{S}_n + \delta \cdot \text{sign}(g_n^S)\|g_n^S\|_*^{q-1}\Upsilon^{1-q}, \tag{4.3}$$

where $g_n^S$ is the gradient of $l_{DH}(\theta; \mathbf{S}_n)$ with respect to $\mathbf{S}_n$ and $\Upsilon$ is defined as $(\frac{1}{N}\sum_{n=1}^N \|g_n^S\|_\star^q)^{1/q}$. To propose an analogous PGD-like algorithm, we iteratively apply (4.3) with adjusted step-size $\beta$ and project the perturbed path back to a ball defined by the constraint in (4.1). During the projection, each sample $\hat{\mathbf{S}}_n$ is updated to

$$\hat{\mathbf{S}}_n \leftarrow \mathbf{S}_n + \max(1, \delta/\text{dist})(\hat{\mathbf{S}}_n - \mathbf{S}_n), \tag{4.4}$$

where $\text{dist} = (\frac{1}{N}\sum_{i=1}^N d(\mathbf{S}_i, \hat{\mathbf{S}}_i)^p)^{1/p}$. The overall procedure, called the Wasserstein Projection Gradient Descent (WPGD), is summarized in Algorithm 1.

---

**Algorithm 1** Wasserstein Projection Gradient Descent (WPGD)

---

1: **for** $i = 1$ to $num\_of\_iteration$ **do**
2:     Compute $\hat{\Upsilon} := (\frac{1}{N}\sum_{n=1}^N \|\nabla_x l_{DH}(\theta; \hat{\mathbf{S}}_n)\|_\star^q)^{1/q}$
3:     **for** $n = 1$ to $N$ **do**
4:         $\hat{\mathbf{S}}_n \leftarrow \hat{\mathbf{S}}_n + \beta \cdot \text{sign}(\nabla_x l_{DH}(\theta; \hat{\mathbf{S}}_n))\|\nabla_x l(\theta; \hat{\mathbf{S}}_n)\|_*^{q-1}\hat{\Upsilon}^{1-q}$,
5:     **end for**
6:     $\hat{\mathbf{S}}_1, \ldots, \hat{\mathbf{S}}_n \leftarrow \text{Proj}_{\hat{B}_\delta(\mu)}(\hat{\mathbf{S}}_1, \ldots, \hat{\mathbf{S}}_n)$
7: **end for**

---

**Wasserstein Budget Projection Gradient Descent (WBPGD).** We can interpret $d(\mathbf{S}_n, \hat{\mathbf{S}}_n)^p$ in (4.1) as a budget allocated to path $\mathbf{S}_n$ during the attack. This intuition motivates a reparameterization of the perturbed sample as

$$\hat{\mathbf{S}}_n = \mathbf{S}_n + \text{budget}_n \times \text{direction}_n, \tag{4.5}$$

where for each sample $\mathbf{S}_n$, the variable $\text{budget}_n \in \mathbb{R}_{\geq 0}$ represents the magnitude of the perturbation, and $\text{direction}_n$, with the same dimension as $\mathbf{S}_n$, denotes its direction bounded within $[-1, 1]$. Then, the optimization problem (4.1) can be considered as allocating $\text{budget}_n$ with the restriction $(\frac{1}{N}\sum_{n=1}^N \text{budget}_n^p)^{1/p} = \delta$ and optimizing the perturbation direction for each sample. We have the following one-step updates on $\text{budget}_n$ and $\text{direction}_n$ based on (4.3).

**Lemma 4.1.** *For the perturbation $\hat{\mathbf{S}}_n$ satisfying* (4.3)*, we have the equivalent representation*

$$\hat{\mathbf{S}}_n = \mathbf{S}_n + \text{budget}_n \times \text{direction}_n. \tag{4.6}$$

*Let $g_n^b$ and $g_n^d$ be the gradient of $l_{DH}(\theta; \mathbf{S}_n + b \times d)$ with respect to $b$ and $d$ when $b = 0$ and $d = \text{sign}(\nabla_S l_{DH}(\theta; \mathbf{S}_n))$, then*

$$\text{budget}_n = \delta \cdot (g_n^b)^{q-1}\Upsilon^{1-q}, \quad \text{direction}_n = \text{sign}(g_n^d). \tag{4.7}$$

*Furthermore, $\Upsilon$ in* (4.3) *satisfies $\Upsilon = (\frac{1}{N}\sum_{n=1}^N (g_n^b)^q)^{1/q}$.*

By applying the above updates iteratively with step size and projection, we propose a new PGD-like algorithm that optimizes the budget allocation and perturbation direction for each sample independently. The full procedure is summarized in Algorithm 2.

---
**Algorithm 2** Wasserstein Budget Projection Gradient Descent (WBPGD)
---
1: **for** $i = 1$ to $num\_of\_iteration$ **do**
2:     Compute $g_n^b$ and $g_n^d$ through back-propagation for $n = 1, \ldots, N$
3:     Compute $\hat{\Upsilon} := (\frac{1}{N} \sum_{n=1}^{N} (g_n^b)^q)^{1/p}$
4:     **for** $n = 1$ to $N$ **do**
5:         $\text{budget}_n \leftarrow \text{budget}_n + \beta \cdot (g_n^b)^{q-1} \hat{\Upsilon}^{1-q}$
6:         $\text{direction}_n \leftarrow \text{direction}_n + \beta/\delta \cdot \text{sign}(g_n^d)$
7:         Clamp $\text{direction}_n$ to $[-1, 1]$
8:         $\hat{\mathbf{S}}_n \leftarrow \mathbf{S}_n + \text{budget}_n \times \text{direction}_n$
9:     **end for**
10:    $\hat{\mathbf{S}}_1, \ldots, \hat{\mathbf{S}}_n \leftarrow \text{Proj}_{\hat{B}_\delta(\mu)}(\hat{\mathbf{S}}_1, \ldots, \hat{\mathbf{S}}_n)$
11: **end for**
---

**Extension to Heston model.** We now consider how to extend the above algorithms to models with several series of data as input, such as the Heston model. In this case, the network strategy uses two input series: the price process $\mathbf{S}$ and the variance process $\mathbf{v}$. During the attack phase, we perturb either the price process $\mathbf{S}$ alone (referred to as S-Attack) or both the price and variance processes simultaneously (referred to as SV-Attack). For S-Attack, we can directly apply the above algorithms. For SV-attack, we first need to define, for a weight $\lambda > 0$, the distance on $(\mathbf{S}, \mathbf{V})$ as

$$d((\mathbf{S}, \mathbf{V}), (\hat{\mathbf{S}}, \hat{\mathbf{v}})) = ((\max_t |S_t - \hat{S}_t|)^p + (\lambda \cdot \max_t |v_t - \hat{v}_t|)^p)^{1/p}. \tag{4.8}$$

The price and variance are weighted differently as they are on different scales. Under this distance, we have $d((\mathbf{S}, \mathbf{v}), (\hat{\mathbf{S}}, \hat{\mathbf{v}}))^p = d(\mathbf{S}, \hat{\mathbf{S}})^p + \lambda^p d(\mathbf{v}, \hat{\mathbf{v}})^p$ and the following result by direct calculation.

**Corollary 4.2.** *Under Assumption 3.1 and Assumption 3.2, for input $x = (\mathbf{S}, \mathbf{v})$ with distance defined as* (4.8)*, $\Upsilon$ becomes*

$$\Upsilon = (\sum_{n=1}^{N} \|\nabla_S l(\theta; \mathbf{S}_n, \mathbf{v}_n)\|_1^q + \|1/\lambda \cdot \nabla_{\mathbf{v}} l(\theta; \mathbf{S}_n, \mathbf{v}_n)\|_1^q)^{1/q} \tag{4.9}$$

*and the perturbation* (3.9) *can be written as*

$$\mathbf{S} \mapsto \mathbf{S} + sign(\nabla_S l(\theta; \mathbf{S}, \mathbf{v}))\|\nabla_x l(\theta; \mathbf{S}, \mathbf{v})\|_*^{q-1} \Upsilon^{1-q} \tag{4.10}$$

$$\mathbf{v} \mapsto \mathbf{v} + 1/\lambda \cdot sign(\nabla_v l(\theta; \mathbf{S}, \mathbf{v}))\|1/\lambda \cdot \nabla_v l(\theta; \mathbf{S}, \mathbf{v})\|_*^{q-1} \Upsilon^{1-q}. \tag{4.11}$$

This corollary reveals that applying an adversarial perturbation jointly to the price and variance processes $(\mathbf{S}, \mathbf{v})$ under the defined metric (4.8) is equivalent to independently perturbing the price series $\mathbf{S}$ and the scaled variance series $\lambda \mathbf{v}$ separately using the $l_\infty$-distance. Consequently, in practical implementation, we can conveniently treat $\mathbf{S}$ and $\lambda \mathbf{v}$ as separate input sequences, by transforming the original sample set $\{\mathbf{S}_1, \ldots, \mathbf{S}_n\}$ into $\{\mathbf{S}_1, \lambda \mathbf{v}_1, \ldots, \mathbf{S}_n, \lambda \mathbf{v}_n\}$.

## 5 Experimental Results

### 5.1 Attacks on classical deep hedging strategies

We start with applying distributional adversarial attacks to neural network strategies trained in the classical deep hedging [2] setting, i.e., on the Heston model with CVaR loss function. As detailed in Section 4, both S-attack and SV-attack are implemented using the WPGD and WBPGD algorithms introduced in Algorithms 1 and 2. The resulting hedging loss across varying perturbation magnitudes $\delta$ is summarized in Table 1. The base case ($\delta = 0$) corresponds to the unperturbed hedging loss. Table 1 shows that the adversarial loss significantly increases as the attack magnitude ($\delta$) increases. Additionally, the WBPGD method consistently outperforms the WPGD method, especially at larger perturbation magnitudes. Therefore, WBPGD will be used as the main attack method in subsequent experiments.

To understand practical implications and quantify the extent of distortion in the path space due to the attack, we assessed perturbation impacts through covariance matrix comparisons, as shown in Table 2. The Frobenius distance between the covariance matrices of perturbed and original paths,

given the original covariance matrix norm ($\approx 386$), indicates minor covariance distortions, especially for small perturbations ($\delta < 0.1$). Specifically, simultaneous perturbations on both price and variance processes (SV-Attack) result in less pronounced covariance changes compared to perturbations only on the price process (S-Attack). In the appendix we also report autocorrelation function comparisons.

Overall, together with with Table 1, these analyses show that adversarial attacks may significantly deteriorate the hedging strategy's performance, even under seemingly modest perturbations, thereby underscoring the necessity for robust neural network models in financial applications.

Table 1: Robustness of classical deep hedging strategy under different attack methods and magnitudes

| $\delta$ | 0 | 0.01 | 0.03 | 0.05 | 0.1 | 0.3 | 0.5 |
|---|---|---|---|---|---|---|---|
| S-WBPGD | 1.9280 | 1.9642 | 2.0432 | 2.1356 | 2.4466 | 4.5771 | 8.0745 |
| SV-WBPGD | 1.9280 | 1.9659 | 2.0485 | 2.1451 | 2.4660 | 4.5898 | 7.7391 |
| S-WPGD | 1.9280 | 1.9642 | 2.0431 | 2.1343 | 2.4369 | 4.5148 | 7.5411 |
| SV-WPGD | 1.9280 | 1.9652 | 2.0457 | 2.1378 | 2.4393 | 4.4802 | 7.4678 |

Table 2: Distance between covariance matrices of perturbed and original paths

| $\delta$ | 0.01 | 0.03 | 0.05 | 0.1 | 0.3 | 0.5 |
|---|---|---|---|---|---|---|
| S-Attack | 0.1421 | 0.4038 | 0.6295 | 1.0248 | 2.3864 | 4.5057 |
| SV-Attack | 0.1355 | 0.3836 | 0.5929 | 0.9548 | 2.1787 | 3.8617 |

## 5.2 Adversarial training

We introduce adversarial training, which aims to enhance the robustness of strategies against distributional adversarial attacks and hence solve the DRO problem. We adopt the standard deep hedging methodology [2] as baseline and expand it to incorporate adversarial examples during training. Experimental details (e.g., network architecture, hyperparameters) are provided in the appendix.

**Loss functions.** In standard deep hedging, the network parameters $\theta$ are optimized using the loss function defined by $\mathcal{L}_{clean}(\theta) = \sum_{n=1}^{N} l_{DH}(\theta; X_n)$. In adversarial training, we separate the optimization problem into two parts: the inner maximization problem and the outer minimization problem. During the inner maximization part, the network parameters $\theta$ are fixed and we apply a distributionally adversarial attack to find the worst-case perturbation. With the adversarial perturbation $\hat{X}_n$ obtained from the distributional adversarial attack, we can then minimize the expected loss function with respect to the network parameters $\theta$ in the outer minimization part. Following [20, 22], our training uses an enhanced loss function

$$\mathcal{L}_{adv}(\theta) = \alpha \cdot \sum_{n=1}^{N} l_{DH}(\theta; X_n) + \sum_{n=1}^{N} l_{DH}(\theta; \hat{X}_n), \tag{5.1}$$

where $\{\hat{X}_1, ..., \hat{X}_n\}$ are adversarially perturbed versions of the original samples and $\alpha$ balances the importance of clean versus adversarial samples. This leads to an iterative process of adversarial training, alternating between generating adversarial samples and optimizing the network parameters.

**Dataset.** We conduct our experiments using three well-established financial models: Black–Scholes, Heston, and the General Affine Diffusion (GAD) model. Here we present results for the Heston model in Section 5.3, while model details and results for the other models are provided in the appendix. Results for real market data are reported in Section 5.4 below. For each model, we generate extensive training datasets of 100,000 sample paths. To examine robustness across varying dataset sizes, we partition each dataset into smaller subsets with sizes $N$ ranging from 5,000 to 100,000 samples. Neural networks are independently trained on these subsets, and the average performance is assessed and reported on a fixed test set, which contains 1 million paths, generated from the same distribution. In addition, we generate a validation set of 100,000 paths, but only $N$ paths will be used for validation, so that the training is exposed to only a limited number of data depending on $N$.

**Robustness of adversarial training.** Strategies trained with our adversarial procedure achieve lower losses under distributional perturbations than classical deep hedging strategies, indicating that

the method effectively approximates the desired distributionally robust solution. Detailed experiments and results are provided in the appendix.

## 5.3 Adversarial training improves out-of-sample and out-of-distribution performance

**Out-of-sample performance.** Given robust strategies alongside the corresponding clean strategies, Figure 1a evaluates hedging performance on a test set of 1 million simulated paths approximating the true data-generating distribution. The midline presents the average performance of strategies trained on partitioned datasets of size $N$, while the shaded area shows the performance range. From the plot, adversarial training significantly outperforms conventional methods, especially when data is scarce: at $N = 5,000$, SV-Attack achieves a 54% lower mean hedging loss than clean training (2.86 vs 6.21) while reducing worst-case outcomes by 66% (max loss 3.60 vs 10.62). When the data size becomes larger, the empirical distribution becomes closer to the true underlying distribution, and all strategies approach near-identical performance ($\sim 1.95$). However, the robust strategy still outperforms the clean one even at relatively smaller scales. In addition, though S-Attack and SV-Attack show comparable average performance, SV-Attack exhibits tighter variance across all $N$, indicating the advantage of allowing perturbation in the variance process.

**Out-of-distribution performance.** In prior experiments, test data were drawn from the same distribution as the training data. To further assess generalizability, we now evaluate the strategy on out-of-distribution (OOD) samples generated under perturbed parameter regimes. Specifically, we generate 100 new parameter configurations by scaling the original values by factors uniformly sampled from $[0.9, 1.1]$, introducing bounded deviations of $\pm 10\%$. For each perturbed configuration, 10,000 sample paths are simulated, resulting in a comprehensive OOD dataset of 1 million paths. Figure 1b illustrates the strategy's performance on this OOD dataset. Despite the models never encountering these perturbed parameter regimes during training, the observed performance trends align closely with the out-of-sample results in Figure 1a, underscoring the robustness of the approach under parameter distribution shifts.

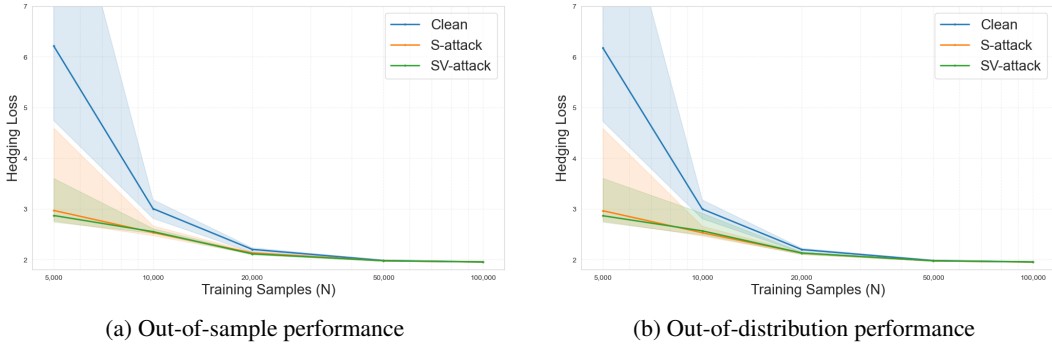

(a) Out-of-sample performance        (b) Out-of-distribution performance

Figure 1: Comparative hedging performance under Heston models. Shaded regions indicate min-max ranges across training partitions.

## 5.4 Experiments on Market Data

In this section, we evaluate the performance of adversarial training on real market data. Specifically, we train hedging strategies using historical daily closing prices from leading companies in the S&P 500 index from [47], covering the period from 26 September 2008 to 8 March 2020. The time period is chosen in line with the benchmark method [11] with which we compare our method.

For this evaluation, we constructed two synthetic datasets based on an additional model introduced in the appendix. The **FIX** dataset is simulated using the General Affine Diffusion (GAD) model with fixed parameters estimated from a 250-day period prior to 8 March 2020. The **ROBUST** dataset, following [11], is also based on the GAD model but incorporates parameter robustness by sampling parameters uniformly from intervals determined by the extreme values across 26 rolling estimates. These estimates are obtained from 250-day windows updated every 100 days between 26 September 2008 and 8 March 2020. For each dataset (FIX and ROBUST), we generate 100,000 paths for training,

validation, and testing. The starting price is set as the closing price of the respective company on 8 March 2020, and each trajectory is scaled to begin from 10.

Directly training on the FIX and ROBUST datasets corresponds to the methods proposed in Deep Hedging [2] and Robust Deep Hedging [11], respectively. Building on this foundation, we implement the adversarial framework described in Section 5.2, resulting in two adversarial variants of the strategies. Similar to [11], we evaluate out-of-sample performance on real data by computing profit and loss (PnL) for hedging strategies on the tested price trajectories starting form 9 March 2020. Table 3 summarizes results for five companies.[2], where bold entries indicate top-performing results per company. To address the limitations of using a single price trajectory, we refer the reader to the Appendix for details of the evaluation protocol and an additional evaluation result.

Several notable patterns emerge from this analysis. The clean strategy trained on the ROBUST dataset generally outperforms the clean strategy trained on the FIX dataset. This aligns with the conclusion of [11], which highlights the benefit of incorporating parameter uncertainty into the training data. However, our proposed adversarial training framework consistently achieves better performance overall. When trained on the FIX dataset, adversarial training yields notable improvements over clean training across all companies. In cases where clean training on the ROBUST dataset performs better than adversarial training on the FIX dataset, applying adversarial training to the ROBUST dataset further enhances performance. These findings underscore the effectiveness of the adversarial training framework we propose in addressing model misspecification and enhancing strategy robustness.

Table 3: Testing performance (P&L) on tested price trajectory

| Method | AAPL | AMZN | BRK-B | GOOGL | MSFT |
|---|---|---|---|---|---|
| Clean training on FIX [2] | -2.261 | -0.981 | -2.086 | -1.275 | -4.341 |
| Clean training on ROBUST [11] | -0.830 | -0.849 | -0.281 | 0.026 | -0.223 |
| Adversarial training on FIX | **-0.579** | **-0.291** | **-0.127** | -0.459 | -0.564 |
| Adversarial training on ROBUST | -0.739 | -0.860 | -0.294 | **0.199** | **-0.144** |

# 6 Conclusion

We presented a robust deep hedging framework that leverages adversarial training under Wasserstein ambiguity to address the challenges of model misspecification and distributional shifts. By formulating the distributionally robust optimization problem as a minimax objective and approximating it through a tractable reformulation, we demonstrated how deep hedging strategies can be trained adversarially. Empirical results across a variety of widely used synthetic models and data regimes show that adversarially trained strategies achieve improved out-of-sample and out-of-distribution performance, especially under structural changes or limited data availability. Further experiments on real market data suggest that these strategies also generalize well beyond simulated settings, maintaining robustness in periods of market stress. These findings underscore the practical value of robust training methods in financial environments characterized by uncertainty and instability.

**Limitations and future work.** Our adversarially trained deep hedging framework enhances robustness, but its performance remains sensitive to the choice of the Wasserstein radius, which may differ across models. Future work could explore potentially stronger distributional adversarial attack methods and explore extensions to alternative ambiguity set geometries beyond Wasserstein balls.

**Acknowledgment** GH's research is supported by the Department of Mathematics at Imperial College London through the Roth Scholarship. We also thank G-Research for the travel support to attend NeurIPS.

---

[2]Including Apple (AAPL), Amazon (AMZN), Microsoft (MSFT), Google (GOOGL), and Berkshire Hathaway (BRK-B)

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

# Appendix

## A    Wasserstein Distance

For a Wasserstein distributionally robust optimization problem (1.1), the ambiguity set $B_\delta(\mu)$ is defined as a Wasserstein ball centered at $\mu \in \mathcal{P}(\mathbb{R}^d)$, i.e.,

$$B_\delta(\mu) = \left\{ \eta \in \mathcal{P}(\mathbb{R}^d) : W_p(\mu, \eta) \leq \delta \right\}. \tag{A.1}$$

For any $p \in (1, \infty)$, the Wasserstein distance $W_p(\mu, \eta)$ between two distributions $\mu$ and $\eta$ is defined as

$$W_p(\mu, \eta) = \Big( \inf_{\gamma \in \Pi(\mu, \eta)} \int_{\mathbb{R}^d \times \mathbb{R}^d} d(x, y)^p d\gamma(x, y) \Big)^{1/p}, \tag{A.2}$$

where $\Pi(\mu, \eta)$ denotes the set of all joint distributions on $\mathbb{R}^d \times \mathbb{R}^d$ with marginals $\mu$ and $\eta$ and $d$ denotes a metric on $\mathbb{R}^d$.

For $p = \infty$, the Wasserstein distance becomes the minimal maximal displacement between two distributions

$$W_\infty(\mu, \eta) = \inf_{\gamma \in \Pi(\mu, \eta)} \{\gamma\text{-ess}\sup d(x, y)\}. \tag{A.3}$$

Here, $\gamma$-ess sup denotes the essential supremum with respect to the measure $\gamma$ over $\mathbb{R}^d \times \mathbb{R}^d$.

## B    Deep Hedging Examples

**Case 1: Black–Scholes Model with Entropic Risk Measure.**    In this scenario, we assume that the asset price follows the classical Black–Scholes model

$$dS_t = mS_t \, dt + \sigma S_t \, dW_t, \tag{B.1}$$

where $m$ is the drift, $\sigma$ is the volatility, and $W_t$ is a standard Brownian motion. The process is discretized in time for training the neural network.

Here, as the process is Markovian, we define the information process directly as the price process $S_t$, which provides all the necessary information to make decisions at time $t$. The network strategy in (2.1) is simplified to

$$\delta_t = f_{\theta_t}(S_t). \tag{B.2}$$

We consider hedging a European call option with terminal payoff

$$P(S_T) = \max(S_T - K, 0), \tag{B.3}$$

where $K$ is the strike price. To account for risk aversion in the objective function, we adopt the entropic risk measure

$$\rho(Z) = \frac{1}{\lambda} \log \mathbb{E}\left[e^{-\lambda Z}\right], \tag{B.4}$$

where $\lambda > 0$ is the risk aversion parameter. This risk measure is commonly used in finance to model the risk preferences of investors [44].

By [2, Example 3.8], the the entropic risk measure admits the OCE form

$$\rho(Z) = \inf_{\omega \in \mathbb{R}} \left\{ \omega + \mathbb{E}\left[ \exp(-\lambda(Z + \omega)) - \frac{1 + \log \lambda}{\lambda} \right] \right\}. \tag{B.5}$$

Moreover, the corresponding optimal $\omega$ in (B.5) is given by

$$\omega^* = \frac{1}{\lambda} \log \mathbb{E}[\lambda \cdot \exp(-\lambda Z)]. \tag{B.6}$$

The corresponding deep hedging loss is then defined as

$$l_{DH}(\theta, \omega, \mathbf{S}) = \omega - \frac{1 + \log \lambda}{\lambda} + \exp(-\lambda(\text{PnL}(\theta, \mathbf{S}) + \omega)). \tag{B.7}$$

**Case 2: Heston Model with Conditional Value-at-Risk (CVaR).** In this scenario, we assume that the asset price follows the Heston stochastic volatility model:

$$dS_t^1 = mS_t^1 \, dt + \sqrt{v_t} S_t^1 \, dW_t^1, \quad dv_t = a(b - v_t) \, dt + \sigma \sqrt{v_t} \, dW_t^2, \tag{B.8}$$

where $v_t$ is the stochastic variance process, and the Brownian motions $W_t^1$ and $W_t^2$ satisfy

$$\mathbb{E}[dW_t^1 dW_t^2] = \rho \, dt. \tag{B.9}$$

The parameters $a$, $b$, and $\sigma$ control the mean reversion speed, long-run variance level, and volatility of volatility, respectively.

As $v_t$ itself is not directly tradable, to hedge the volatility risk, we introduce a second price process representing a variance swap corresponding to the tradable asset.

The variance swap at time $t$ is given by

$$S_t^2 = \int_0^t v_s \, ds + \frac{v_t - b}{a}(1 - e^{-a(T-t)}) + b(T - t), \tag{B.10}$$

where the integral is approximated by the trapezium rule in practice.

We then hedge through trading both the underlying asset and the variance swap, i.e., we define the combined price process as $S_t = (S_t^1, S_t^2)$. Moreover, since the network needs both the price of the underlying and variance to make decisions, the information process is defined as $I_t = (S_t^1, v_t)$. The Heston model is Markovian with respect to this information process, so the network strategy becomes:

$$\delta_t = f_{\theta_t}(S_t^1, v_t). \tag{B.11}$$

We again consider hedging a European call option with the same terminal payoff as in (B.3).

To evaluate hedging performance under downside risk, we adopt the Conditional Value-at-Risk (CVaR) risk measure at confidence level $\alpha \in [0, 1)$

$$\text{CVaR}_\alpha(Z) = \frac{1}{1 - \alpha} \int_0^{1-\alpha} \text{VaR}_\gamma(Z) d\gamma \tag{B.12}$$

$$\text{VaR}_\gamma(Z) = \inf \{z \in \mathbb{R} \, : \, \mathbb{P}(Z \leq -z) < \gamma\}. \tag{B.13}$$

This risk measure captures the expected loss in the worst $1 - \alpha$ fraction of outcomes and is widely used in risk management [48]. The CVaR can be written in OCE form

$$\text{CVaR}_\alpha(Z) = \inf_{\omega \in \mathbb{R}} \left\{ \omega + \frac{1}{1 - \alpha} \mathbb{E}[\max(-Z - \omega, 0)] \right\}, \tag{B.14}$$

where the optimal $\omega$ is attained at the $\alpha$-quantile of $Z$.

We then define the corresponding deep hedging loss as

$$l_{DH}(\theta, \omega, \mathbf{S}^1, \mathbf{v}) = \omega + \frac{1}{1 - \alpha} \max(-\text{PnL}(\theta, \mathbf{S}^1, \mathbf{v}) - \omega, 0). \tag{B.15}$$

**Case 3: General Affine Diffusion (GAD) Model with Entropic Risk Measure.** In this scenario, we assume the asset price follows a General Affine Diffusion (GAD) process:

$$dS_t = (b_0 + b_1 S_t) \, dt + (a_0 + a_1 S_t)^\gamma \, dW_t, \tag{B.16}$$

where $W_t$ is a standard Brownian motion and the parameters $b_0$, $b_1$, $a_0$, $a_1$, and $\gamma$ control the drift and diffusion characteristics.

To discretize the process for numerical implementation, we apply the Euler–Maruyama scheme. Following the robust approach of [11], we introduce parameter uncertainty through intervals, where at each path and each time step, parameters are drawn uniformly from their respective intervals:

$$\begin{aligned}
&\text{for } t = 1, \ldots, T: \\
&\quad \Delta W_t \sim \mathcal{N}(0, \Delta t), \\
&\quad a_0 \sim U[\underline{a}_0, \overline{a}_0], \quad a_1 \sim U[\underline{a}_1, \overline{a}_1], \quad b_0 \sim U[\underline{b}_0, \overline{b}_0], \quad b_1 \sim U[\underline{b}_1, \overline{b}_1], \\
&\quad S_t = S_{t-1} + (b_0 + b_1 S_{t-1})\Delta t + (a_0 + a_1 S_{t-1})^\gamma \Delta W_t.
\end{aligned} \tag{B.17}$$

By taking intervals as a point value, the above process approximates the classical GAD process.

As in the Black–Scholes model, the network strategy is defined using only the current asset price:

$$\delta_t = f_{\theta_t}(S_t). \tag{B.18}$$

Following the same setup as in [11], we consider hedging an Asian at-the-money put option with the terminal payoff

$$P(\mathbf{S}) = \max\left(S_0 - \frac{1}{T}\sum_{t=1}^{T} S_t, 0\right). \tag{B.19}$$

We use the same entropic risk measure as in the Black-Scholes model (see Case 1 above).

## C  Proofs

*Proof of Theorem 3.3.* Our proof follows the approach of [23, Theorem 4.1], adapted to the classification setting. Specifically, we build on [46, Theorem 2.1] and its proof, which we restate below as a theorem.

**Theorem C.1** (Adapted from [46, 23]). *Under Assumption 3.1 and Assumption 3.2, the following statements hold.*

(i) *The first-order sensitivity expansion as $\delta \downarrow 0$ ensures*

$$V(\delta) = V(0) + \delta\Upsilon + o(\delta), \quad where \quad \Upsilon := \mathbb{E}_{x\sim\mu}\left[\|\nabla_x l(\theta; x)\|_*^q\right]^{1/q} \tag{C.1}$$

*and $q$ is the conjugate exponent of $p$, satisfying $1/q + 1/p = 1$.*

(ii) *Furthermore, $V(\delta)$ can be approximated by*

$$V(\delta) = \mathbb{E}_{\eta_\delta}[l(\theta, x)] + o(\delta) \quad as \quad \delta \downarrow 0 \tag{C.2}$$

*where the perturbed distribution $\eta_\delta$ is explicitly given by*

$$\eta_\delta = \left[x \mapsto x + \delta \cdot h(\nabla_x l(\theta; x))\|\nabla_x l(\theta; x)\|_*^{q-1}\Upsilon^{1-q}\right]_{\#}\mu. \tag{C.3}$$

In the data-driven framework we choose $\mu = \frac{1}{N}\sum_{n=1}^{N}\delta_{X_n}$. Then $\Upsilon$ in (C.1) becomes the average

$$\Upsilon = (\frac{1}{N}\sum_{n=1}^{N}\|\nabla_x l(\theta; X_n)\|_*^q)^{1/q} \tag{C.4}$$

and the perturbation $\eta_\delta$ in (C.3) becomes a uniform distribution on the perturbed dataset $\{\hat{X}_1, \ldots, \hat{X}_N\}$, where each $\hat{X}_n$ satisfies

$$\hat{X}_n = X_n + \delta \cdot h(\nabla_x l(\theta; X_n))\|\nabla_x l(\theta; X_n)\|_*^{q-1}\Upsilon^{1-q}. \tag{C.5}$$

$\square$

*Proof of Lemma 3.4.* By Theorem 3.3, $\eta_\delta$ is of the form $\frac{1}{N}\sum_{n=1}^{N}\delta_{\hat{X}_n}$ and satisfy

$$\begin{aligned}
\frac{1}{N}\sum_{i=1}^{N} d(X_i, \hat{X}_i)^p &= \frac{1}{N}\sum_{i=1}^{N}\left\|\delta \cdot h(\nabla_x l(\theta; X_n))\|\nabla_x l(\theta; X_n)\|_*^{q-1}\Upsilon^{1-q}\right\|^p \\
&= \frac{\delta^p}{\Upsilon^{(q-1)p}} \cdot \frac{1}{N}\sum_{i=1}^{N}\|h(\nabla_x l(\theta; X_n))\|\|\nabla_x l(\theta; X_n)\|_*^{(q-1)p} \\
&= \frac{\delta^p}{\Upsilon^q} \cdot \frac{1}{N}\sum_{i=1}^{N}\|\nabla_x l(\theta; X_n)\|_*^q \\
&= \delta^p
\end{aligned} \tag{C.6}$$

where we use $\|h(x)\| = \sup_{\|x\|_*\leq 1}\langle h(x), x\rangle = \sup_{\|x\|_*\leq 1}\|x\|_* = 1$, $(q-1)p = (1-1/q)pq = q$ for exponent conjugate and (C.4). Therefore, we have $\eta_\delta \in \hat{B}_\delta(\mu)$.

Moreover, for $\mu = \frac{1}{N} \sum_{n=1}^{N} \delta_{X_n}$ and $\hat{\mu} = \frac{1}{N} \sum_{n=1}^{N} \delta_{\hat{X}_n}$, we have

$$\pi = \frac{1}{N} \sum_{n=1}^{N} \delta_{(X_n, \hat{X}_n)} \in \Pi(\mu, \hat{\mu}). \tag{C.7}$$

Therefore, by the definition of the Wasserstein distance,

$$W_p(\mu, \hat{\mu}) \le (\int d(x, y)^p \, d\pi(x, y))^{1/p} = (\frac{1}{N} \sum_{i=1}^{N} d(X_i, \hat{X}_i)^p)^{1/p}. \tag{C.8}$$

If $(\frac{1}{N} \sum_{i=1}^{N} d(X_i, \hat{X}_i)^p)^{1/p} < \delta$, so is $W_p(\mu, \hat{\mu})$, hence $\hat{B}_\delta(\mu) \subset B_\delta(\mu)$.

Overall, we proved that $\eta_\delta \in \hat{B}_\delta(\mu) \subset B_\delta(\mu)$. Therefore, $\mathbb{E}_{\eta_\delta}[l(\theta; x)] \le V_\theta^e(\delta) \le V_\theta(\delta)$ and

$$0 \le \frac{1}{\delta}(V_\theta(\delta) - V_\theta^e(\delta)) \le \frac{1}{\delta}(V_\theta(\delta) - \mathbb{E}_{\eta_\delta}[l(\theta; x)]). \tag{C.9}$$

By Theorem 3.3, RHS $\to 0$ as $\delta \to 0$, so $\frac{1}{\delta}(V_\theta(\delta) - V_\theta^e(\delta))$ also converges to 0. In other words,

$$V_\theta(\delta) = V_\theta^e(\delta) + o(\delta) \quad \text{as } \delta \downarrow 0. \tag{C.10}$$

$\square$

*Proof of Lemma 4.1.* For $g_n^S = \nabla_{\mathbf{S}} l(\theta; \mathbf{S}_n)$, the updates (4.3) can be separated into updates on $\text{budget}_n$ and $\text{direction}_n$

$$\Upsilon = (\frac{1}{N} \sum_{n=1}^{N} \|g_n^S\|_*)^{1/q}, \quad \text{budget}_n = \|g_n^S\|_*^{q-1} \Upsilon^{1-q}, \quad \text{direction}_n = \text{sign}(g_n^S). \tag{C.11}$$

By the chain rule, we have $g_n^b = \langle g_n^S, \text{direction}_n \rangle = \|g_n^S\|_*$ and $g_n^d = g_n^S / \text{budget}_n$. The update above becomes

$$\Upsilon = (\frac{1}{N} \sum_{n=1}^{N} (g_n^b)^q)^{1/q}, \quad \text{budget}_n = (g_n^b)^{q-1} \Upsilon^{1-q}, \quad \text{direction}_n = \text{sign}(g_n^d), \tag{C.12}$$

which proves the lemma. $\square$

Corollary 4.2 is a special case of the following more general result, which we will prove next.

**Corollary C.2.** *Consider the setting of Theorem 3.3 with inputs of the form $x = (x^1, ..., x^d) \in \mathbb{R}^{d \times (T+1)}$ representing d-dimensional sequences of length $T + 1$. Set the norm as*

$$\|x\| := (\sum_{i=1}^{d} (\lambda_i \|x^i\|_\infty)^p)^{1/p}, \tag{C.13}$$

*where $\| \cdot \|_\infty$ is the infinity norm defined in the space of the trajectories $\mathbb{R}^{T+1}$. Over the input samples $\{X_1, \dots, X_N\}$, where each sample contains d trajectories $X_n = (X_n^1, \dots, X_n^d)$, we define $g_n = (g_n^1, \dots, g_n^d) = \nabla_x l(\theta; (X_n^1, \dots, X_n^d))$ to be the gradient with respect to the input. In this setting, the perturbation (3.9) can be written as*

$$\hat{X}_n^i = X_n^i + \frac{1}{\lambda_i} \text{sign}(g_n^i) \|\frac{1}{\lambda_i} g_n^i\|_1^{q-1} \Upsilon^{1-q} \tag{C.14}$$

*for $i = 1, \dots, d$ and $n = 1, \dots, N$. Moreover, $\Upsilon$ becomes*

$$\Upsilon = (\sum_{n=1}^{N} \sum_{i=1}^{d} \|\frac{1}{\lambda_i} g_n^i\|_1^q)^{1/q} \tag{C.15}$$

*where $\| \cdot \|_1$ is the $l_1$-norm defined on $\mathbb{R}^{T+1}$.*

*Proof.* We first show that the norm in (C.13) has dual norm defined as

$$\|y\|_* = (\sum_{i=1}^{d} (\frac{1}{\lambda_i} \|y^i\|_1)^q)^{1/q}. \tag{C.16}$$

With the standard pairing $\langle x, y \rangle = \sum_i \langle x^i, y^i \rangle$, we estimate

$$\left| \langle x, y \rangle \right| \leq \sum_{i=1}^{d} \|x^i\|_\infty \|y^i\|_1 \leq \big( \sum_{i=1}^{d} (\lambda_i \|x^i\|_\infty)^p \big)^{1/p} \big( \sum_{i=1}^{d} (\tfrac{1}{\lambda_i} \|y^i\|_1)^q \big)^{1/q} = \|x\| \, \|y\|_*, \quad \text{(C.17)}$$

where the inequalities hold by Hölder's inequality.

By setting $\lambda_i \|x^i\|_\infty \propto \frac{1}{\lambda_i} \|y^i\|_1$ and $x^i = \|x^i\|_\infty \mathrm{sign}(y^i)$ for each $i$, for any $y$ there exists $x$ such that $\|x\| = 1$ and $\left| \langle x, y \rangle \right| = \|x\| \, \|y\|_*$. Hence (C.16) indeed defines the dual norm by definition. The corresponding function $h(y)$ such that $\langle h(y), y \rangle = y$ is defined as

$$h(y) = \frac{1}{\|y\|_*^{q-1}} \big( \tfrac{1}{\lambda_1} \mathrm{sign}(y^i) \| \tfrac{1}{\lambda_1} y^i \|_1^{q-1}, \ldots, \tfrac{1}{\lambda_d} \mathrm{sign}(y^d) \| \tfrac{1}{\lambda_d} y^d \|_1^{q-1} \big). \quad \text{(C.18)}$$

Recall that the perturbation in (3.9) is

$$\hat{X}_n = X_n + \delta \cdot h(g_n) \|g_n\|_*^{q-1} \Upsilon^{1-q}. \quad \text{(C.19)}$$

By (C.18), $h(g_n)$ becomes,

$$h(g_n) = \frac{1}{\|g_n\|_*^{q-1}} \big( \tfrac{1}{\lambda_1} \mathrm{sign}(g_n^1) \| \tfrac{1}{\lambda_1} g_n^1 \|_1^{q-1}, \ldots, \tfrac{1}{\lambda_d} \mathrm{sign}(g_n^d) \| \tfrac{1}{\lambda_d} g_n^d \|_1^{q-1} \big). \quad \text{(C.20)}$$

Therefore, bringing (C.20) into (C.19), each trajectory in each sample $X_n$ is perturbed to

$$\hat{X}_n^i = X_n^i + \delta \cdot \big( \tfrac{1}{\|g_n\|_*^{q-1}} \tfrac{1}{\lambda_i} \mathrm{sign}(g_n^i) \| \tfrac{1}{\lambda_i} g_n^i \|_1^{q-1} \big) \cdot \|g_n\|_*^{q-1} \Upsilon^{1-q}$$

$$= X_n^i + \frac{\delta}{\lambda_i} \mathrm{sign}(g_n^i) \| \tfrac{1}{\lambda_i} g_n^i \|_1^{q-1} \Upsilon^{1-q} \quad \text{(C.21)}$$

for $i = 1, \ldots, d$ and $n = 1, \ldots, N$. Finally,

$$\Upsilon = \big( \sum_{n=1}^{N} \|g_n\|_*^q \big)^{1/q} = \big( \sum_{n=1}^{N} \sum_{i=1}^{d} \| \tfrac{1}{\lambda_i} g_n^i \|_1^q \big)^{1/q}. \quad \text{(C.22)}$$

$\square$

## D  Experimental Details

Here we provide additional experimental details regarding the adversarial training introduced in Section 5.2. Readers can refer to the code for a comprehensive implementation.

**Network Architecture.**   The neural network architecture remains consistent with the standard deep hedging framework [2], characterized by decision-making at each time step $t$ through:

$$\delta_t = f_t^{\theta_t}(I_t), \quad \text{(D.1)}$$

where $I_t$ encapsulates all relevant information available at step $t$. In line with [2], each $f_t^{\theta_t}$ comprises two hidden layers, each with 20 neurons, batch normalization, and ReLU activation.

**Training Procedure.**   Our training procedure begins with a preliminary phase of clean training to establish stable initial parameters. Specifically, this phase lasts 100 epochs for the BS model and 300 epochs for the more complex Heston model. Subsequently, the network undergoes adversarial training for an additional 200 epochs (BS) or 400 epochs (Heston), alternating adversarial example generation and optimization of Eq. (5.1). For comparison, we train baseline networks (clean strategies) exclusively with clean training for an equivalent total duration (300 epochs for BS, 700 epochs for Heston).

**Optimizer and Learning rate.**   Optimization utilizes the Adam optimizer, with decaying learning rate—initially set to 0.005 for BS and 0.05 for Heston. The batch size is set to 10,000 unless the dataset size $N$ is smaller, in which case the entire dataset is utilized per batch.

**Hyperparameters.**  Critical adversarial training hyperparameters include $\alpha$, tested at $0, 1, 10$ to gauge the relative influence of clean versus adversarial loss, and perturbation magnitude $\delta$, explored across $\{0.001, 0.003, 0.005, 0.01, 0.03, 0.05, 0.1, 0.3, 0.5\}$. Hyperparameter selection is performed by evaluating performance on a validation set of size $N$ and selecting the hyperparameters yielding the best validation results.

**Adversarial attack.**  During the experiment, we employ the WBPGD algorithm detailed in Algorithm 2 for adversarial attacks. We execute this algorithm for 20 iterations, setting the step-size as $\beta = \frac{4}{20}\delta$, which is dependent on the perturbation magnitude $\delta$. Additionally, for the two models considered, input trajectories have identical initial values across all samples. Consequently, we avoid perturbing the initial values by explicitly setting both the perturbation and corresponding gradient components to zero.

**Computation time.**  All computational runs are conducted without GPU on AMD EPYC 7742 or Intel Icelake Xeon Platinum 8358 processors equipped with less than 64GB of memory. For standard adversarial training involving 100,000 sample paths, the computation time is approximately 3 hours for the Black-Scholes model and around 10 hours for the Heston model. In contrast, classical deep hedging without adversarial training requires roughly one-tenth of this computational effort. The increased computational demand for adversarial training is reasonable, as each network update includes an additional 20 iterations of adversarial perturbations, enhancing the network's robustness.

**Cash-invariant property of convex risk measure.**  By the cash-invariance property [44] of the convex risk measure $\rho$, we have $\rho(Z + c) = \rho(Z) - c$ for any random variable $Z$ and constant $c$ (representing cash injection). Therefore, in practice, we directly set $p_0$ in (2.2) to 0 as it does not affect the optimization problem. Note that we are only interested in hedging here; for pricing the an appropriate choice of $p_0$ could be determined as described in [2].

# E  Additional Experimental Results

In this section, we provide supplementary experimental results to further validate and contextualize the analyses presented in Section 5.

## E.1  Autocorrelation function comparison

Building upon the analysis presented in Section 5.1, we further examine the impact of adversarial perturbations by comparing the autocorrelation functions (ACFs) of the adversarially perturbed trajectories against those of the original trajectories. For a path $\{x_t\}$, the ACF is defined as:

$$\mathrm{ACF}(x, \mathrm{lag}) = \frac{1}{\sigma^2} \sum_{i=0}^{\mathrm{lag}} \frac{1}{N-i} \sum_{t=1}^{N-i} (x_t - \bar{x})(x_{t+i} - \bar{x}), \qquad (\mathrm{E.1})$$

where $\bar{x}$ is the empirical mean of the path $x$, and $\sigma^2$ is its empirical variance.

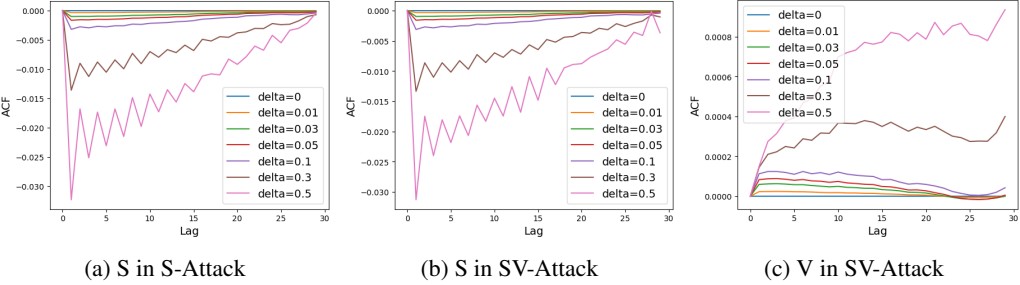

| (a) S in S-Attack | (b) S in SV-Attack | (c) V in SV-Attack |

Figure 2: Difference in Autocorrelation function (ACF) of perturbed paths for different $\delta$ values and original paths.

Fig. 2 illustrates the difference in ACF of perturbed paths and original paths. It reveals minimal deviations ($< 0.005$) in autocorrelation for perturbations with magnitude $\delta < 0.1$. Despite these

modest ACF differences, the corresponding hedging errors are notably large (see Table 1), thereby undermining further that an adversarially perturbed sample path distribution may lead to significant hedging losses, even though comparisons of ACF and covariance matrices suggest that the perturbed distribution is close to the original distribution.

## E.2 Robustness of adversarial training

For the robust strategy trained using adversarial training detailed in Section 5.2, we examine the adversarial loss to confirm training effectiveness. We evaluate robust strategies trained under distributional adversarial attacks with perturbation magnitudes $\delta = 0.01$ and $0.1$, comparing their performance against a clean strategy trained on an identical dataset of 100,000 samples. Table 4 illustrates the adversarial loss for each strategy under varying perturbation levels. The robust strategies consistently yield significantly lower adversarial losses compared to the clean strategy, clearly demonstrating that our robust training framework successfully provides a computationally tractable approximation of the theoretically optimal distributionally robust optimization (DRO) solution, enhancing model resilience against adversarial perturbations. Additionally, there is an expected trade-off observed in the loss: for the robust strategy trained with $\delta = 0.1$, the loss at no or small perturbation levels is higher, but the adversarial loss at larger perturbation magnitudes is substantially reduced, aligning with the designed training objective.

Table 4: Comparison of robust and clean strategies under different perturbation magnitudes

| $\delta$ | 0 | 0.01 | 0.03 | 0.05 | 0.1 | 0.3 | 0.5 |
|---|---|---|---|---|---|---|---|
| clean | 1.9162 | 1.9598 | 2.0530 | 2.1560 | 2.4659 | 4.6069 | 8.0379 |
| robust ($\delta = 0.01$) | 1.9186 | 1.9541 | 2.0298 | 2.1130 | 2.3675 | 4.1626 | 7.1445 |
| robust ($\delta = 0.1$) | 1.9796 | 2.0035 | 2.0534 | 2.1064 | 2.2549 | 3.1161 | 4.3156 |

## E.3 Results on adversarial training

**Optimal hyperparameter choice.**    As detailed in Section 3 and Appendix D, our robust training framework introduces two critical hyperparameters: the attack radius $\delta$ controlling perturbation magnitude, and the balance weight $\alpha$ modulating between nominal and adversarial losses. Through grid search across $(\delta, \alpha) \in [0.001, 0.5] \times \{0, 1, 10\}$, we identify optimal configurations that maximize validation performance for each training set size $N$. The optimal hyperparameters for the Heston model are presented in Table 5a where we can observe that the optimal $\delta$ decreases from $0.5$ at $N = 5,000$ to $0.005$ at $N = 100,000$. This phenomenon arises because smaller training sizes induce greater divergence between the empirical distribution $\mu_N$ and the true data-generating distribution $\mu$, thus larger adversarial perturbations ($\delta$) are required to bridge this distributional gap. The optimal parameters for the Black-Scholes model show a similar pattern, see Table 5b.

**Detailed Heston results.**    Table 6 provides detailed out-of-sample performance results, supplementing the information shown in Figure 1a. Specifically, the table shows improvements in robust strategy performance as the sample size becomes large and demonstrates that the SV-Attack strategy exhibits lower variance compared to the S-Attack strategy.

**Black-Scholes results.**    Figure 3 and Table 7 show comprehensive results for the Black-Scholes model, revealing patterns analogous to the Heston model. However, the gap between robust and clean strategies is relatively smaller than in Heston. This aligns with expectations - the simpler Black-Scholes model offers fewer exploitable gaps for adversarial training to mitigate, particularly in volatility dynamics.

Table 5: Hyperparameter Choices for Heston and Black-Scholes Models

(a) Heston Model Hyperparameters

| Training Samples (N) | S-Attack $\delta$ | S-Attack $\alpha$ | SV-Attack $\delta$ | SV-Attack $\alpha$ |
|---|---|---|---|---|
| 5,000 | 0.3 | 1 | 0.5 | 1 |
| 10,000 | 0.1 | 10 | 1.0 | 10 |
| 20,000 | 0.05 | 1 | 0.1 | 1 |
| 50,000 | 0.03 | 0 | 0.03 | 0 |
| 100,000 | 0.01 | 0 | 0.005 | 0 |

(b) Black-Scholes Model Hyperparameters

| Training Samples (N) | $\delta$ | $\alpha$ |
|---|---|---|
| 5,000 | 0.01 | 10 |
| 10,000 | 0.005 | 10 |
| 20,000 | 0.003 | 10 |
| 50,000 | 0.001 | 1 |
| 100,000 | 0.001 | 0 |

Table 6: Detailed out-of-sample performance across sample sizes (N) for SV-Attack, S-Attack, and Clean strategies on Heston model

| Strategy | N | Avg Loss | Min Loss | Max Loss | Variance |
|---|---|---|---|---|---|
| SV-Attack | 5,000 | 2.8644 | 2.7386 | 3.5975 | 0.0346 |
| SV-Attack | 10,000 | 2.5460 | 2.5173 | 2.6087 | 0.0008 |
| SV-Attack | 20,000 | 2.1063 | 2.0928 | 2.1282 | 0.0002 |
| SV-Attack | 50,000 | 1.9706 | 1.9669 | 1.9742 | 2.6e-5 |
| SV-Attack | 100,000 | 1.9469 | 1.9469 | 1.9469 | – |
| S-Attack | 5,000 | 2.9646 | 2.7605 | 4.5912 | 0.1574 |
| S-Attack | 10,000 | 2.5287 | 2.4694 | 2.6629 | 0.0028 |
| S-Attack | 20,000 | 2.1259 | 2.1027 | 2.1489 | 0.0004 |
| S-Attack | 50,000 | 1.9705 | 1.9665 | 1.9745 | 3.2e-5 |
| S-Attack | 100,000 | 1.9472 | 1.9472 | 1.9472 | – |
| Clean | 5,000 | 6.2095 | 4.7379 | 10.6187 | 2.0887 |
| Clean | 10,000 | 3.0000 | 2.8068 | 3.1755 | 0.0129 |
| Clean | 20,000 | 2.1955 | 2.1329 | 2.2266 | 0.0014 |
| Clean | 50,000 | 1.9773 | 1.9705 | 1.9841 | 9.2e-5 |
| Clean | 100,000 | 1.9503 | 1.9503 | 1.9503 | – |

Table 7: Detailed BS model performance metrics across sample sizes (N) for robust and Clean strategies

| Strategy | N | Avg Loss | Min Loss | Max Loss |
|---|---|---|---|---|
| Robust | 5,000 | 2.4109 | 2.4040 | 2.4195 |
| Robust | 10,000 | 2.3947 | 2.3920 | 2.3976 |
| Robust | 20,000 | 2.3855 | 2.3852 | 2.3861 |
| Robust | 50,000 | 2.3798 | 2.3794 | 2.3802 |
| Robust | 100,000 | 2.3769 | 2.3769 | 2.3769 |
| Clean | 5,000 | 2.4136 | 2.4055 | 2.4208 |
| Clean | 10,000 | 2.3976 | 2.3947 | 2.3993 |
| Clean | 20,000 | 2.3860 | 2.3854 | 2.3866 |
| Clean | 50,000 | 2.3798 | 2.3794 | 2.3801 |
| Clean | 100,000 | 2.3772 | 2.3772 | 2.3772 |

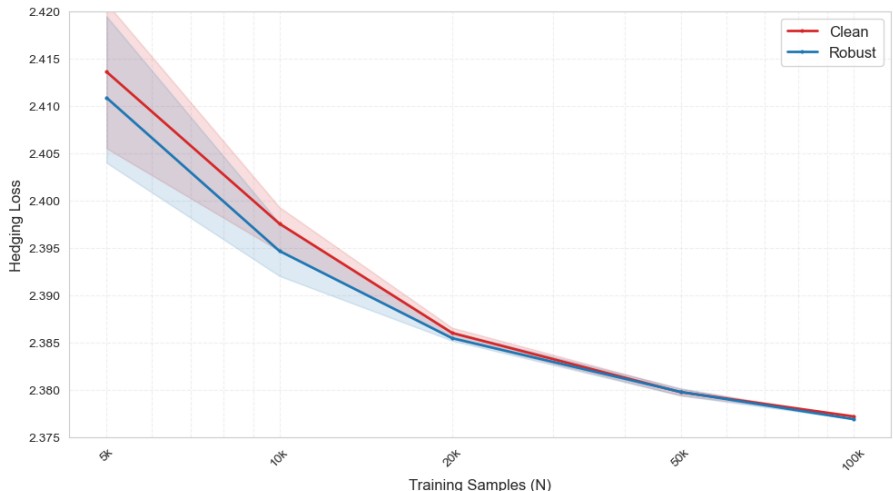

Figure 3: Out-of-sample hedging performance comparison under Black-Scholes dynamics, comparing robust training with clean strategies. Shaded regions indicate min-max ranges across training partitions.

## E.4 Results on Heston models with transaction costs

In this section, we provide an additional example demonstrating the effectiveness of adversarial training by considering the Heston model with transaction costs. Following Equation (2.2), we modify the profit and loss (P&L) by incorporating a transaction cost term, which is proportional to the value of the assets traded:

$$\text{PnL}(\theta, \mathbf{I}) = p_0 + \sum_{t=1}^{T} \left( \delta_t^\top(\theta, \mathbf{I})(S_{t+1}(\mathbf{I}) - S_t(\mathbf{I})) + \epsilon S_t^\top(\mathbf{I}) |\delta_t(\theta, \mathbf{I}) - \delta_{t-1}(\theta, \mathbf{I})| \right) - P(S_T(\mathbf{I}))$$

where $\epsilon$ denotes the transaction cost rate, which we set to 0.005 in this experiment.

Apart from adding an additional transaction cost term to the loss function, the experimental setup follows the same procedure described in Sections 5.2 and Appendix D. To account for transaction costs, we also consider a recurrent neural network architecture, in which the portfolio holding at each time step is passed as input to the next. In this setting, the update rule in (B.11) becomes

$$\delta_t = f_{\theta_t}(S_t^1, v_t, \delta_{t-1}). \tag{E.2}$$

We refer to the original feedforward network as NetSim, and the recurrent version as NetRec.

As in Figure 1, we report the out-of-sample performance of both adversarial and clean training in Figure 4 and Figure 5 on NetSim and NetRec, respectively. The results show that the adversarially trained strategy consistently outperforms the clean strategy. However, this behavior differs from the case without transaction costs: instead of plateauing as the sample size increases, the hedging loss decreases more rapidly. We also observe that although NetRec is more complex, its performance does not outperform NetSim. Both findings may indicate that even with 100,000 samples, the data does not fully capture the underlying distribution in the more complex setting. Under such conditions, adversarial training demonstrates clear advantages over clean training. These findings further highlight the potential of adversarial methods to enhance robustness in data-limited or distributionally uncertain environments.

## E.5 Results on real market data

In Section 5.4, we evaluated performance on a single real price trajectory, following the approach of [11]. However, relying on a single path leads to the absence of validation data and exposure to training randomness. Hence, we adopt the following evaluation protocol. For adversarial training, we use $N = 100{,}000$ training samples and report the average out-of-sample performance across six

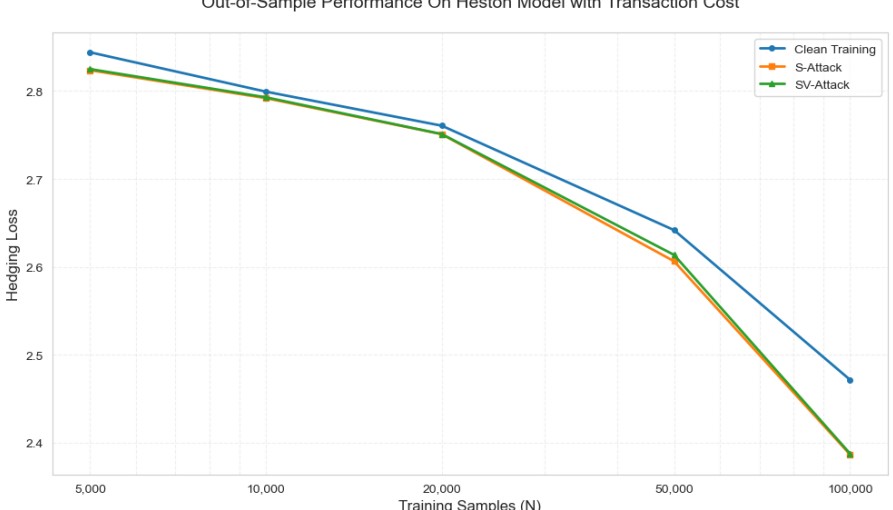

Figure 4: Out-of-sample hedging performance comparison under Heston model with transaction cost on NetSim, comparing among clean strategies and robust strategies under S-attack and SV-attack.

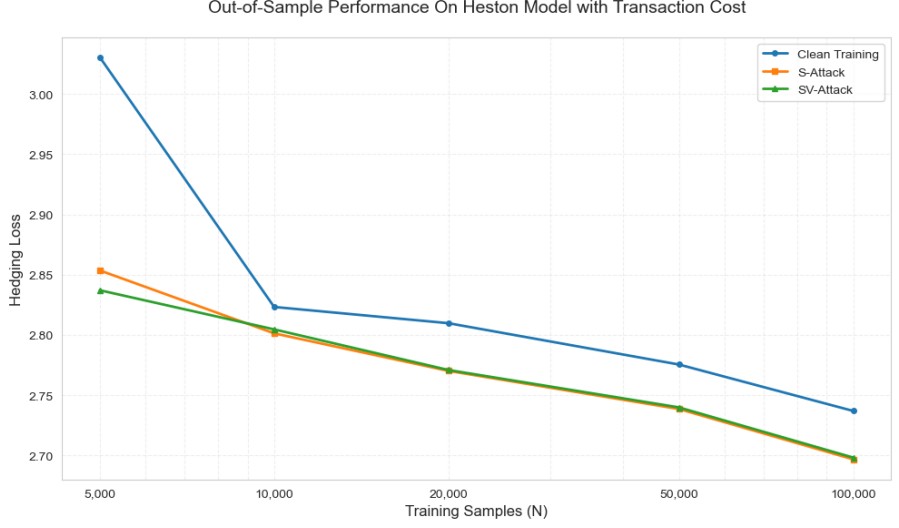

Figure 5: Out-of-sample hedging performance comparison under Heston model with transaction cost on NetRec, comparing among clean strategies and robust strategies under S-attack and SV-attack.

strategies corresponding to $(\alpha, \delta) \in \{0, 1\} \times \{0.03, 0.05, 0.1\}$. For clean training, we average the results over three independent runs with identical settings.

For a more thorough assessment of the model performance, we construct a more comprehensive REAL dataset by applying a rolling window of length 30 over the period from 7 March 2020 to 30 September 2021. Each path is normalized to start at a price of 10, yielding a total of 300 sample trajectories. We adopt the entropic risk measure to evaluate strategy performance, thereby avoiding the need to compute option prices—one of the key motivations for using this loss function. Performance results are summarized in Table 8, where lower values indicate better outcomes, and bold entries highlight the best performance in each column. We observe that the adversarial strategy trained on the FIX dataset consistently outperforms all others across all stocks. In contrast, clean training on the ROBUST dataset does not offer notable improvements over FIX dataset, and applying adversarial training to the ROBUST dataset further degrades performance. These findings suggest that explicitly incorporating robust parameter intervals into the data generation process may be unnecessary—or even counterproductive. Since adversarial training already mitigates the impact of model misspecification, the use of parameter intervals estimated from historical data that may no longer reflect current market conditions can introduce outdated constraints, ultimately diminishing the relevance and effectiveness of the strategy.

Table 8: Testing performance (entropic loss on REAL dataset)

| Method | AAPL | AMZN | BRK-B | GOOGL | MSFT |
|---|---|---|---|---|---|
| Clean training on FIX [2] | 0.2920 | 0.2971 | 0.1993 | 0.2608 | 0.3269 |
| Clean training on ROBUST [11] | 0.3053 | 0.3005 | 0.2410 | 0.3797 | 0.2692 |
| Adversarial training on FIX | **0.2517** | **0.2408** | **0.1692** | **0.2401** | **0.1891** |
| Adversarial training on ROBUST | 0.3077 | 0.3148 | 0.2437 | 0.3965 | 0.2776 |

Finally, we complement that analysis by reporting the performance on the FIX and ROBUST datasets in Tables 9 and 10, respectively. Similar to Black-Scholes and Heston model, neural networks are independently trained on subsets of size $N$, and we compute the average performance on the test set. For each adversarial training result, the hyperparameters, including the attack radius $\delta$ and the balance weight $\alpha$, are selected using a validation set of size $N$.

When models are trained and evaluated on the FIX dataset, we observe trends consistent with those seen in the Black-Scholes and Heston experiments: adversarial training consistently outperforms clean training, with the advantage especially pronounced when the sample size is small, suggesting particular effectiveness in data-scarce regimes. The ROBUST dataset, which generalizes the FIX dataset by incorporating model misspecification, further highlights the benefits of adversarial training—strategies trained adversarially on FIX significantly outperform their clean-trained counterparts, demonstrating enhanced robustness to distributional shifts. When models are trained directly on the ROBUST dataset, the pattern becomes more nuanced: performance does not always improve with larger sample sizes, yet adversarial training still generally provides a performance edge. Together, these results underscore the practical value of adversarial training in improving generalization under complex and uncertain market conditions.

## F   Extension to other models

In Section 3, we introduced distributional adversarial attack algorithms specifically for the Black-Scholes model, with input $\mathbf{I} = \mathbf{S}$, and the Heston model, with input $\mathbf{I} = (\mathbf{S}^1, \mathbf{v})$. In this section, we generalize these approaches to an arbitrary model characterized by input trajectories $\mathbf{I} = (\mathbf{I}^1, \ldots, \mathbf{I}^d)$.

We begin by defining a distance measure on the general input space

$$d(\mathbf{I}, \hat{\mathbf{I}}) = \left( \sum_{i=1}^{d} (\lambda_i \cdot \max_t |I_t^i - \hat{I}_t^i|)^p \right)^{1/p}, \tag{F.1}$$

analogous to the distance measure introduced for the Heston model in (4.8). This definition allows distinct perturbation scales for each trajectory, consistent with the Heston model framework.

Table 9: Performance comparison on FIX dataset

| Company | Method | N=5000 | 10000 | 20000 | 50000 | 100000 |
|---------|--------|--------|-------|-------|-------|--------|
| *Trained on FIX dataset* | | | | | | |
| AAPL | Adversarial | 0.1737 | 0.1752 | 0.1706 | 0.1667 | 0.1643 |
| | Clean | 0.1960 | 0.1943 | 0.1782 | 0.1734 | 0.1745 |
| | Improvement | **0.0223** | **0.0191** | **0.0076** | **0.0067** | **0.0101** |
| AMZN | Adversarial | 0.1927 | 0.1785 | 0.1763 | 0.1753 | 0.1737 |
| | Clean | 0.2401 | 0.1970 | 0.1867 | 0.1782 | 0.1770 |
| | Improvement | **0.0474** | **0.0185** | **0.0104** | **0.0029** | **0.0033** |
| BRK-B | Adversarial | 0.1412 | 0.1368 | 0.1353 | 0.1353 | 0.1352 |
| | Clean | 0.1867 | 0.1649 | 0.1523 | 0.1408 | 0.1363 |
| | Improvement | **0.0455** | **0.0281** | **0.0169** | **0.0055** | **0.0011** |
| GOOGL | Adversarial | 0.2258 | 0.2025 | 0.1985 | 0.1981 | 0.1976 |
| | Clean | 0.2955 | 0.2224 | 0.2065 | 0.2005 | 0.1984 |
| | Improvement | **0.0696** | **0.0199** | **0.0080** | **0.0023** | **0.0008** |
| MSFT | Adversarial | 0.1549 | 0.1491 | 0.1431 | 0.1436 | 0.1418 |
| | Clean | 0.1878 | 0.1789 | 0.1551 | 0.1479 | 0.1473 |
| | Improvement | **0.0329** | **0.0298** | **0.0120** | **0.0043** | **0.0055** |

Utilizing Corollary C.2, a direct generalization of Corollary 4.2, the update rule for each scaled trajectory is given by:

$$\lambda_i \hat{\mathbf{I}}_n^i = \lambda_i \mathbf{I}_n^i + \delta \cdot \text{sign}(\frac{1}{\lambda_i} g_n^i) \| \frac{1}{\lambda_i} g_n^i \|_1^{q-1} \Upsilon^{1-q}, \tag{F.2}$$

which closely parallels the update step for the complete sample set

$$\hat{\mathbf{I}}_n = \mathbf{I}_n + \delta \cdot h(g_n) \|g_n\|_*^{q-1} \Upsilon^{1-q}. \tag{F.3}$$

Furthermore, the total distance $\sum_{n=1}^N d(\mathbf{I}_n, \hat{\mathbf{I}}_n)^p$ and the term $\Upsilon^p$ used during the iterative update process can naturally be decomposed into sums across both trajectory dimensions ($i = 1, \ldots, d$) and individual samples ($n = 1, \ldots, N$).

Consequently, similar to the approach in the Heston model, each trajectory will be independently perturbed according to an $l_\infty$-norm metric, effectively transforming the original set of samples into a structured set of scaled trajectories $\{\lambda_i \mathbf{I}_n^i\}_{n=1,\ldots,N}^{i=1,\ldots,d}$.

Table 10: Performance comparison on ROBUST dataset

| Company | Method | N=5000 | 10000 | 20000 | 50000 | 100000 |
|---|---|---|---|---|---|---|
| *Trained on FIX dataset* | | | | | | |
| AAPL | Adversarial | 0.4346 | 0.5023 | 0.4748 | 0.4404 | 0.4490 |
| | Clean | 0.4387 | 0.5354 | 0.5679 | 0.5328 | 0.4831 |
| | Improvement | **0.0041** | **0.0331** | **0.0931** | **0.0924** | **0.0341** |
| AMZN | Adversarial | 0.4983 | 0.5342 | 0.5875 | 0.5958 | 0.5994 |
| | Clean | 0.5859 | 0.6509 | 0.6677 | 0.6743 | 0.6737 |
| | Improvement | **0.0876** | **0.1167** | **0.0802** | **0.0785** | **0.0743** |
| BRK-B | Adversarial | 0.5548 | 0.5330 | 0.5416 | 0.4872 | 0.5029 |
| | Clean | 0.6408 | 0.6733 | 0.6876 | 0.6441 | 0.6351 |
| | Improvement | **0.0860** | **0.1403** | **0.1459** | **0.1569** | **0.1322** |
| GOOGL | Adversarial | 0.5276 | 0.6236 | 0.6563 | 0.6054 | 0.6055 |
| | Clean | 0.5378 | 0.6707 | 0.6814 | 0.6775 | 0.6844 |
| | Improvement | **0.0102** | **0.0471** | **0.0252** | **0.0721** | **0.0788** |
| MSFT | Adversarial | 0.5859 | 0.6357 | 0.5716 | 0.5505 | 0.5678 |
| | Clean | 0.6358 | 0.6870 | 0.6993 | 0.6847 | 0.7124 |
| | Improvement | **0.0499** | **0.0513** | **0.1276** | **0.1342** | **0.1446** |
| *Trained on ROBUST dataset* | | | | | | |
| AAPL | Adversarial | 0.3936 | 0.3607 | 0.3463 | 0.3703 | 0.4325 |
| | Clean | 0.4229 | 0.3705 | 0.3699 | 0.3875 | 0.4510 |
| | Improvement | **0.0293** | **0.0098** | **0.0237** | **0.0172** | **0.0185** |
| AMZN | Adversarial | 0.4276 | 0.3525 | 0.2701 | 0.2401 | 0.2470 |
| | Clean | 0.4505 | 0.3627 | 0.2633 | 0.2647 | 0.3065 |
| | Improvement | **0.0229** | **0.0102** | -0.0068 | **0.0246** | **0.0595** |
| BRK-B | Adversarial | 0.3702 | 0.2849 | 0.1912 | 0.2123 | 0.2467 |
| | Clean | 0.4117 | 0.3618 | 0.2265 | 0.1800 | 0.3076 |
| | Improvement | **0.0415** | **0.0769** | **0.0354** | -0.0323 | **0.0609** |
| GOOGL | Adversarial | 0.3748 | 0.2584 | 0.1434 | 0.0985 | 0.1427 |
| | Clean | 0.4222 | 0.3351 | 0.2100 | 0.1276 | 0.2207 |
| | Improvement | **0.0474** | **0.0766** | **0.0667** | **0.0291** | **0.0780** |
| MSFT | Adversarial | 0.3885 | 0.2800 | 0.1945 | 0.2170 | 0.2399 |
| | Clean | 0.4115 | 0.3473 | 0.2535 | 0.2674 | 0.2944 |
| | Improvement | **0.0230** | **0.0673** | **0.0590** | **0.0504** | **0.0544** |

