# OpenReview forum: "Distributional Adversarial Attacks and Training in Deep Hedging"
_NeurIPS.cc/2025/Conference — NeurIPS 2025 poster_

### Official Review · Reviewer_sQgP · 2025-06-29

**Clarity:** 2
**Significance:** 3
**Originality:** 3
**Rating:** 4
**Confidence:** 3

**Summary:**

This paper investigates the robustness of classical deep hedging strategies under distributional shifts using the concept of adversarial attacks. The authors demonstrate that standard deep hedging models are highly vulnerable to small perturbations in the input distribution, leading to significant performance degradation. To address this, they propose an adversarial training framework tailored to increase the robustness of deep hedging strategies.

**Questions:**

See Weaknesses.

I think the idea of enhancing robustness through distributional adversarial attacks is innovative and holds potential for practical applications in financial risk management.However, the paper lacks a coherent flow, making it challenging for readers to follow the logical progression of ideas. Additionally,the paper would benefit from a more detailed discussion on the limitations of existing methods and how the proposed approach overcomes these limitations.

**Ethical Concerns:**

["NO or VERY MINOR ethics concerns only"]

**Final Justification:**

Thanks for the authors' responses. The rebuttal has addressed most of my concerns. The quality of the article will be further improved based on the author's current feedback.

As stated in my initial evaluation, this paper introduces a Wasserstein framework that combines distributional adversarial training with deep hedging. The topic is meaningful and this work might be the first to address the robustness of deep hedging strategies under distributional shifts using adversarial training.

Considering that the technical contribution outweighs other minor issues (like writing and experiments)，I keep my original positive rating.

**Quality:**

3

**Strengths And Weaknesses:**

Strengths:

1. The paper introduces a novel framework that combines distributional adversarial training with deep hedging. This might be the first work to address the robustness of deep hedging strategies under distributional shifts using adversarial training.

2. The paper develops specific algorithms (e.g., Wasserstein Projection Gradient Descent and Wasserstein Budget Projection Gradient Descent) tailored for distributional adversarial attacks in the context of deep hedging. These algorithms are well-designed and computationally efficient.

3. The paper includes extensive numerical experiments that demonstrate the effectiveness of the proposed adversarial training framework. The results show consistent outperformance of adversarially trained strategies over classical methods in terms of out-of-sample and out-of-distribution performance.

Weaknesses:

1. While the paper presents an interesting approach to enhancing the robustness of deep hedging strategies using adversarial training, I have significant concerns regarding the writing and structure of the paper. Specifically, the paper lacks a literature review or related work section, which is crucial for providing context and situating the work within the broader field. This omission makes it difficult for AI conferences and readers, especially those not deeply familiar with the specific domain, to understand the background, the development of the field, and the unique contributions of this work. Additionally, the paper does not include a conclusion section, which is essential for summarizing the findings and highlighting the significance of the research.

2. The experiments primarily rely on simulated data from well-known financial models (e.g., Heston and Black-Scholes). While these models are widely used, they may not fully capture the complexities of real-world financial markets. The paper could benefit from additional experiments using real-world financial data to further validate the robustness and effectiveness of the proposed methods.

---

> ### Author Rebuttal · Authors · 2025-07-30
>
> Dear reviewer,
>
> Thank you for your time and effort in reviewing our paper and for your fruitful suggestions.
>
> To start with, we would like to share 2 new experiments conducted after our submission.
>
> **Experiment 1: GAD process with real-market data**
>
> In the paper [11], the authors investigate hedging strategies trained on synthetic data generated from a generalized affine diffusion (GAD) process and evaluated on real market prices of leading S&P 500 companies during the COVID-19 crash in March 2020. Following their setup, we construct two training datasets: FIX, where model parameters are fixed based on historical pre-COVID estimates, and ROB, where parameters are sampled from intervals inferred from the same historical data to incorporate model uncertainty. Clean and adversarial training are applied to both datasets, using the entropic risk measure the same as in the Black-Scholes model. As there is no validation set to choose hyperparameters, for adversarial training, we fix the sample size to $N=100,000$ and take the average across six $(α, δ)$ pairs. We provide the average P\&L(eq2.2) below.
>
> | Method| Dataset | AAPL| AMZN| BRK-B| GOOGL| MSFT|
> |-|-|-|-|-|-|-|
> | Clean| FIX| -2.261  | -0.981  | -2.086  | -1.275  | -4.341  |
> | Clean| ROB| -0.830  | -0.849  | -0.281  | 0.026   | -0.223  |
> | Adv  | FIX| **-0.579**  | **-0.291**  | **-0.127**  | -0.459  | -0.564  |
> | Adv  | ROB| -0.739  | -0.860  | -0.294  | **0.199**  | **-0.144**  |
>
> The results show that clean training on the ROB dataset outperforms FIX, confirming the findings of [11]. More importantly, adversarial training consistently improves performance across all settings. Even when ROB-clean outperforms FIX-clean, adversarial training on ROB yields further gains. This highlights the strength of adversarial training in improving robustness under model misspecification.
>
> This setup captures not only the distributional shift caused by market volatility but also the mismatch between assumed model dynamics and real-world behavior, offering a meaningful test of robustness. Our results demonstrate that adversarially trained strategies outperform standard training in this challenging real-market scenario, supporting the practical value of adversarial training methods we proposed.
>
> **Experiment 2: Heston model with transaction cost**
>
> In addition, we extend our study to the Heston model with transaction costs by incorporating an additional cost term into the loss function. The experimental setup otherwise remains identical to the cost-free case. As shown in the following table, like in Fig.1, the adversarially trained strategy consistently outperforms the clean one across all sample sizes. Interestingly, unlike the case without transaction costs—where performance plateaus beyond a certain data size—the hedging loss here continues to decline as the number of training samples increases. This suggests that even with 100,000 samples, the data may not fully represent the true distribution. In such scenarios, adversarial training provides a clear advantage, underscoring its potential to improve robustness under data scarcity or distributional uncertainty.
>
> | N      | clean  | S-attack | SV-attack |
> |-|-|-|-|
> | 5000   | 2.8441 | 2.8237   | 2.8249    |
> | 10000  | 2.7993 | 2.7919   | 2.7929    |
> | 20000  | 2.7606 | 2.7510   | 2.7511    |
> | 50000  | 2.6419 | 2.6064   | 2.6137    |
> | 100000 | 2.4719 | 2.3869   | 2.3880    |
>
> Next, we would like to address your comments one by one.
>
> **Weakness**
>
> >...the paper lacks a literature review or related work section...the paper does not include a conclusion section.
>
> We thank the reviewer for this valuable observation. We fully agree that clearly situating our work within the broader literature is important, especially for readers unfamiliar with the intersection of deep hedging and adversarial robustness.
>
> In Section 1 of our original submission, we integrated a discussion of related works throughout the Introduction (lines 31–36, 54–61) and also included a dedicated "Further related work" paragraph at the end of the section. However, we acknowledge that this scattered format may have made it difficult to appreciate the progression and relevance of these works.
>
> To address this, we have revised the manuscript to include a dedicated and more detailed Related Work section, which systematically:
>
> + Reviews the evolution of deep hedging from its inception [2] to recent methodological extensions. In particular, we now highlight [11] as a key example of addressing parameter misspecification within the deep hedging framework.
>
> + Discusses distributionally robust optimization (DRO) frameworks, particularly those based on Wasserstein metrics [13, 17, 18].
>
> + Connects adversarial training literatures like [19-22] from classical machine learning to the distributional setting in our financial context.
>
> We believe this revision significantly improves accessibility and contextual clarity for readers from both the finance and AI communities.
>
> Motivated by the reviewer's comment about the missing conclusion, we will add the following conclusion to our revised paper:
>
> *We presented a robust deep hedging framework that leverages adversarial training under Wasserstein ambiguity to address the challenges of model misspecification and distributional shifts. By formulating the distributionally robust optimization problem as a minimax objective and approximating it through a tractable reformulation, we demonstrated how deep hedging strategies can be trained adversarially. Empirical results across a variety of widely used synthetic models and data regimes show that adversarially trained strategies achieve improved out-of-sample and out-of-distribution performance, especially under structural changes or limited data availability. Further experiments on real market data suggest that these strategies also generalize well beyond simulated settings, maintaining robustness in periods of market stress. These findings underscore the practical value of robust training methods in financial environments characterized by uncertainty and instability.*
>
> >The experiments primarily rely on simulated data from well-known financial models (e.g., Heston and Black-Scholes). While these models are widely used, they may not fully capture the complexities of real-world financial markets...
>
> We agree with the reviewer that real-world evaluation is essential. The added experiment directly addresses this concern. As shown in **Experiment 1**, adversarially trained strategies continue to outperform standard training even under real market conditions. We believe this experiment complements the synthetic benchmarks and provides further evidence of the practical applicability of our approach.
>
> **Question**
> >...the paper lacks a coherent flow...Additionally, the paper would benefit from a more detailed discussion on the limitations of existing methods and how the proposed approach overcomes these limitations.
>
> Following your advice in the previous comment, we will add a more detailed Related Work section as well as a Conclusion section (see our proposed conclusion above). In addition, since the final version allows for an extra page, we plan to include two new experiments based on real-world market data. We believe these additions will improve the overall coherence of the paper and help readers follow the logical progression of our contributions more easily.
>
> Regarding the limitations of existing methods, we highlight in Section 5.1 how classical deep hedging strategies are vulnerable to adversarial attacks, underscoring the practical need for improved robustness. We will also include a clearer discussion comparing our approach to [11]. While [11] enhances robustness by modifying the model parametrically, its effectiveness remains highly dependent on the specific parameterization chosen. In contrast, our method focuses on distributional robustness, operating over a Wasserstein ball around the empirical distribution without relying on any parametric model structure. This enables our approach to handle distributional shifts more broadly and provides robustness that is independent of model specification.

---

> ### Comment · Reviewer_sQgP · 2025-08-01
> **Rebuttal Acknowledgments**
>
> Thanks for the authors' responses. The rebuttal has addressed most of my concerns.  The quality of the article will be further improved based on the author's current feedback.
>
> As stated in my initial evaluation, this paper introduces a Wasserstein
>  framework that combines distributional adversarial training with deep hedging. The topic is meaningful and this work might be the first to address the robustness of deep hedging strategies under distributional shifts using adversarial training.
>
> Considering that the technical contribution outweighs other minor issues (like writing and experiments)，I keep my original positive rating.

---

> > ### Author Response · Authors · 2025-08-02
> > **Reply Rebuttal Comment by Authors**
> >
> > Thank you for acknowledging our response. We’re pleased that most of the concerns have been clarified. We also appreciate your insistence on a positive rating of our paper.

---

### Official Review · Reviewer_CwkA · 2025-07-01

**Clarity:** 3
**Significance:** 2
**Originality:** 3
**Rating:** 3
**Confidence:** 3

**Summary:**

This paper investigates the robustness of deep hedging strategies against distributional shifts by developing adversarial attacks and training methods. It demonstrates that classical deep hedging models are vulnerable to small perturbations in the input distribution, leading to significant performance degradation. It uses pointwise adversarial attacks to the distributional setting using Wasserstein balls. The approach includes two novel algorithms (WPGD and WBPGD) for generating adversarial distributions and shows empirical improvements in out-of-sample and out-of-distribution performance on Black-Scholes and Heston models.

**Questions:**

1. How do the proposed algorithms scale with the number of time steps T and the dimensionality of the input space? What are the computational complexity bounds?
2. How should practitioners choose the Wasserstein radius in real applications? Is there a principled way to select this parameter based on market characteristics?
3. How sensitive are the results to the choice of neural network architecture? Would the conclusions hold for more modern architectures (e.g., transformers, LSTMs)?
4. Do adversarial examples generated against one hedging model transfer to other models? This would have implications for robustness evaluation.

**Ethical Concerns:**

["NO or VERY MINOR ethics concerns only"]

**Limitations:**

While the paper seems interesting, my concern is that the paper focus on a niche area (ML-quantitative finance). The impact of the work for ML community may not be significant.

**Quality:**

2

**Strengths And Weaknesses:**

# Strengths

- The paper seems to be novel in the sense of systematically applying distributional adversarial attacks to deep hedging.
- The paper provides theoretical grounding by extending sensitivity analysis results from Wasserstein DRO to the deep hedging setting.
- The proposed algorithms (WPGD/WBPGD) offer computationally efficient approximations to the theoretically intractable Wasserstein DRO problem through clever reformulations.

# Weaknesses

- The evaluation is restricted to only two classical financial models (Black-Scholes and Heston). More complex or realistic market models would strengthen the claims.
- All experiments use simulated data from parametric models. Validation on real market data would be more convincing for practical applications.
- While the paper claims computational tractability, there's insufficient analysis of the computational cost compared to standard deep hedging, especially for large-scale problems.

---

> ### Author Rebuttal · Authors · 2025-07-30
>
> Dear reviewer,
>
> Thank you for your time and effort in reviewing our paper and for your fruitful suggestions.
>
> To start with, we would like to share 2 new experiments conducted after our submission.
>
> **Experiment 1: GAD process with real-market data**
>
> In the paper [11], the authors investigate hedging strategies trained on synthetic data generated from a generalized affine diffusion (GAD) process and evaluated on real market prices of leading S&P 500 companies during the COVID-19 crash in March 2020. Following their setup, we construct two training datasets: FIX, where model parameters are fixed based on historical pre-COVID estimates, and ROB, where parameters are sampled from intervals inferred from the same historical data to incorporate model uncertainty. Clean and adversarial training are applied to both datasets, using the entropic risk measure the same as in the Black-Scholes model. As there is no validation set to choose hyperparameters, for adversarial training, we fix the sample size to $N=100,000$ and take the average across six $(α, δ)$ pairs. We provide the average P\&L(eq2.2) below.
>
> | Method| Dataset | AAPL| AMZN| BRK-B| GOOGL| MSFT|
> |-|-|-|-|-|-|-|
> | Clean| FIX| -2.261  | -0.981  | -2.086  | -1.275  | -4.341  |
> | Clean| ROB| -0.830  | -0.849  | -0.281  | 0.026   | -0.223  |
> | Adv  | FIX| **-0.579**  | **-0.291**  | **-0.127**  | -0.459  | -0.564  |
> | Adv  | ROB| -0.739  | -0.860  | -0.294  | **0.199**  | **-0.144**  |
>
> The results show that clean training on the ROB dataset outperforms FIX, confirming the findings of [11]. More importantly, adversarial training consistently improves performance across all settings. Even when ROB-clean outperforms FIX-clean, adversarial training on ROB yields further gains. This highlights the strength of adversarial training in improving robustness under model misspecification.
>
> This setup captures not only the distributional shift caused by market volatility but also the mismatch between assumed model dynamics and real-world behavior, offering a meaningful test of robustness. Our results demonstrate that adversarially trained strategies outperform standard training in this challenging real-market scenario, supporting the practical value of adversarial training methods we proposed.
>
> **Experiment 2: Heston model with transaction cost**
>
> In addition, we extend our study to the Heston model with transaction costs by incorporating an additional cost term into the loss function. The experimental setup otherwise remains identical to the cost-free case. As shown in the following table, like in Fig.1, the adversarially trained strategy consistently outperforms the clean one across all sample sizes. Interestingly, unlike the case without transaction costs—where performance plateaus beyond a certain data size—the hedging loss here continues to decline as the number of training samples increases. This suggests that even with 100,000 samples, the data may not fully represent the true distribution. In such scenarios, adversarial training provides a clear advantage, underscoring its potential to improve robustness under data scarcity or distributional uncertainty.
>
> | N      | clean  | S-attack | SV-attack |
> |-|-|-|-|
> | 5000   | 2.8441 | 2.8237   | 2.8249    |
> | 10000  | 2.7993 | 2.7919   | 2.7929    |
> | 20000  | 2.7606 | 2.7510   | 2.7511    |
> | 50000  | 2.6419 | 2.6064   | 2.6137    |
> | 100000 | 2.4719 | 2.3869   | 2.3880    |
>
> Next, we would like to address your comments one by one.
>
> **Weakness 1**
>
> We selected the Black-Scholes and Heston models to establish a clear and interpretable benchmark aligned with prior work in the deep hedging literature. We conducted additional experiments after submission, including both real-world market data and more complex hedging scenarios.
>
> **Weakness 2**
>
> As described at the start, **Experiment 1** directly addresses this question by training on pre-COVID historical data and evaluating on real market data from March 2020, hence capturing realistic distribution shifts.
>
> **Weakness 3**
>
> We appreciate the reviewer’s concern regarding computational cost. Details on training time and hardware setup are provided in Appendix D. In summary, adversarial training is approximately 10 times more computationally demanding than standard deep hedging, due to the additional inner optimization steps during the adversarial attacks required at each update. While this increase is notable, we believe it is acceptable given the intended use case. Deep hedging is typically applied in risk management settings where models are trained offline, and hedging decisions are executed in real time using a relatively small network. Importantly, the execution time of a trained strategy is comparable regardless of whether it was trained adversarially or classically. Nevertheless, we recognize that improving training efficiency for large-scale or high-frequency retraining scenarios remains a valuable direction for future research.
>
> **Questions 1**
>
> To the best of our knowledge, formal computational complexity bounds are not available even for classical deep hedging, and deriving such bounds remains an open problem in the literature. As a result, scalability is typically assessed through empirical performance.
>
> To illustrate this, we report the average training time per epoch (in seconds) under varying sequence length T and input dimensions using the Black-Scholes model:
>
> Clean Training
> | Input Dim | T = 30 | T = 60 | T = 90 |
> |-|-|-| -|
> | 1             | 0.683  | 0.9968 | 1.4033  |
> | 2             | 0.6769 | 1.0883 | 1.4580   |
> | 4             | 0.6825 | 1.0753 | 1.4703  |
>
> Adversarial Training
> | Input Dim | T = 30 | T = 60  | T = 90 |
> |-|-|-| -|
> | 1             | 6.7717 | 11.5758 | 17.1915 |
> | 2             | 6.513  | 12.1548 | 16.9313 |
> | 4             | 6.2556 | 11.5872 | 17.0486 |
>
> These results indicate that the runtime for training scales reasonably with increasing sequence length T for both clean and adversarial training. In our experiments, the input dimension does not affect the runtime, likely due to the underlying GPU architecture utilizing efficient tensor operation parallelization techniques.
>
> **Questions 2**
>
> In our experiments on simulated data, we performed a grid search over a range of hyperparameter values, including the Wasserstein radius, and selected parameters based on validation performance. However, in the real market experiment introduced above, we did not have access to a dedicated validation set. Instead, we report results averaged over a fixed set of hyperparameter configurations, which helps avoid post-selection bias.
>
> In general, selecting the Wasserstein radius in distributionally robust optimization (DRO) is a well-known challenge and remains an open research problem. To our knowledge, there is no widely accepted principled method for determining this parameter based solely on observable market characteristics (see, e.g., [13]). As shown in Table 4, we do not observe a consistent pattern in the optimal radius across different models, though there is a general trend that overly large radii may degrade performance. Developing a principled selection procedure through statistical uncertainty or market volatility is a valuable direction for future work.
>
> **Questions 3**
>
> In this work, we adopt a standard RNN-based architecture, which is widely used in the deep hedging literature and aligns naturally with the sequential structure of hedging problems. Our empirical results suggest that this architecture is sufficient to capture the relevant temporal dependencies and deliver strong performance in the settings considered.
>
> While it would certainly be interesting to explore more modern architectures such as LSTMs or transformers—particularly in higher-dimensional or more complex market environments—doing so would introduce additional considerations related to model capacity, regularization, and hyperparameter tuning. These aspects, while important, are somewhat beyond the main focus of this work. As our primary goal is to investigate robustness under distributional shifts, we believe the key conclusions can be expected to generalize to other architectures.
>
> **Questions 4**
>
> In our current work, we focus on generating adversarial perturbations tailored to the model being trained, and do not evaluate whether these perturbations also degrade the performance of other independently trained models. This is an intriguing direction, and investigating the transferability of adversarial examples in the hedging context could indeed offer valuable insights into the nature of distributional vulnerabilities and robustness. We see this as an interesting avenue for future research.
>
> **Limitation**
>
> We thank the reviewer for finding our work interesting. While we agree that the paper primarily addresses questions relevant to quantitative finance, the methods and techniques we develop are not limited to that domain. More broadly, our work contributes to the field of robust decision-making based on time-series data, particularly in low-data regimes. This class of problems is rich and general, encompassing stochastic control problems that arise prominently in engineering (e.g., robotics), systems biology, and economics, among other areas.
>
> The distributionally robust optimization (DRO) framework we employ to model adversarial robustness has gained significant attention in the machine learning community over the past decade. However, most existing DRO approaches are confined to the i.i.d. setting and assume access to large amounts of data—for instance, in image classification tasks. In contrast, our work addresses the largely unexplored regime of time-series data with limited training samples. From this broader perspective, we demonstrate that DRO-based modeling can yield substantial performance improvements in such settings.

---

> ### Author Response · Authors · 2025-08-07
>
> Thank you again for your review. We wanted to kindly follow up on the rebuttal we provided. If possible, we would greatly appreciate any feedback you may have at this stage. If there are any remaining questions or points you’d like to discuss, we’d be very happy to provide further clarification.
>
> We sincerely appreciate your time and feedback.

---

### Official Review · Reviewer_P7ou · 2025-07-03

**Clarity:** 2
**Significance:** 3
**Originality:** 3
**Rating:** 4
**Confidence:** 1

**Summary:**

This paper tackles "robust deep hedging" under realistic market uncertainties by utilizing the concept of adversarial robustness. The authors build on the idea of distributional shift by designing Wasserstein-based adversarial attacks in the distributional space, and propose an adversarial training framework that aims to enhance the robustness of hedging strategies.

**Questions:**

While the authors simulate OOD settings via controlled distributional shifts (e.g., Figure 1), I wonder whether real-world market uncertainty—such as the COVID-19 crisis or post-2008 volatility—could be used as a more realistic evaluation. For example, training on pre-crisis data and testing on crisis-period data could provide further insight into the robustness of the proposed method under naturally occurring distribution shifts.

**Ethical Concerns:**

["NO or VERY MINOR ethics concerns only"]

**Final Justification:**

See comments. I keep my rating.

**Limitations:**

See weakness and questions.

**Paper Formatting Concerns:**

No issue.

**Quality:**

3

**Strengths And Weaknesses:**

## Strength

- It is highly interesting how the concept of adverarial training can be utilized in the practical setting.
- The proposed method is based on distributional attack, which is more general than point-wise perturbation and more relevant to real-world risk

## Weakness.

-The number of experiments is limited, and the evaluation seems insufficient to verify the practical applicability of the proposed method.

- It is unclear how well the OOD experiment in Figure 1 reflects real-world financial data. The data seems to be generated based on simple randomness from a uniform distribution, which may not be sufficient to represent realistic market uncertainties.

-  Figure 3 (In the appendix) shows that when the number of training samples is sufficiently large, there is little to no performance gap between robust training and clean training. This suggests that the benefit of robust training mainly appears in low-data regimes. Therefore, it raises questions about the practical effectiveness of the proposed method when ample training data is available.

## Comments.

- As I have limited domain knowledge in finance, I would like to carefully consider the opinions of other reviewers.

- The idea of applying the concept of adversarial robustness to financial engineering is highly interesting.

---

> ### Author Rebuttal · Authors · 2025-07-30
>
> Dear reviewer,
>
> Thank you for your time and effort in reviewing our paper and for your fruitful suggestions.
>
> To start with, we would like to share 2 new experiments conducted after our submission.
>
> **Experiment 1: GAD process with real-market data**
>
> In the paper [11], the authors investigate hedging strategies trained on synthetic data generated from a generalized affine diffusion (GAD) process and evaluated on real market prices of leading S&P 500 companies during the COVID-19 crash in March 2020. Following their setup, we construct two training datasets: FIX, where model parameters are fixed based on historical pre-COVID estimates, and ROB, where parameters are sampled from intervals inferred from the same historical data to incorporate model uncertainty. Clean and adversarial training are applied to both datasets, using the entropic risk measure the same as in the Black-Scholes model. As there is no validation set to choose hyperparameters, for adversarial training, we fix the sample size to $N=100,000$ and take the average across six $(α, δ)$ pairs. We provide the average P\&L(eq2.2) below.
>
> | Method| Dataset | AAPL| AMZN| BRK-B| GOOGL| MSFT|
> |-|-|-|-|-|-|-|
> | Clean| FIX| -2.261  | -0.981  | -2.086  | -1.275  | -4.341  |
> | Clean| ROB| -0.830  | -0.849  | -0.281  | 0.026   | -0.223  |
> | Adv  | FIX| **-0.579**  | **-0.291**  | **-0.127**  | -0.459  | -0.564  |
> | Adv  | ROB| -0.739  | -0.860  | -0.294  | **0.199**  | **-0.144**  |
>
> The results show that clean training on the ROB dataset outperforms FIX, confirming the findings of [11]. More importantly, adversarial training consistently improves performance across all settings. Even when ROB-clean outperforms FIX-clean, adversarial training on ROB yields further gains. This highlights the strength of adversarial training in improving robustness under model misspecification.
>
> This setup captures not only the distributional shift caused by market volatility but also the mismatch between assumed model dynamics and real-world behavior, offering a meaningful test of robustness. Our results demonstrate that adversarially trained strategies outperform standard training in this challenging real-market scenario, supporting the practical value of adversarial training methods we proposed.
>
> **Experiment 2: Heston model with transaction cost**
>
> In addition, we extend our study to the Heston model with transaction costs by incorporating an additional cost term into the loss function. The experimental setup otherwise remains identical to the cost-free case. As shown in the following table, like in Fig.1, the adversarially trained strategy consistently outperforms the clean one across all sample sizes. Interestingly, unlike the case without transaction costs—where performance plateaus beyond a certain data size—the hedging loss here continues to decline as the number of training samples increases. This suggests that even with 100,000 samples, the data may not fully represent the true distribution. In such scenarios, adversarial training provides a clear advantage, underscoring its potential to improve robustness under data scarcity or distributional uncertainty.
>
> | N      | clean  | S-attack | SV-attack |
> |-|-|-|-|
> | 5000   | 2.8441 | 2.8237   | 2.8249    |
> | 10000  | 2.7993 | 2.7919   | 2.7929    |
> | 20000  | 2.7606 | 2.7510   | 2.7511    |
> | 50000  | 2.6419 | 2.6064   | 2.6137    |
> | 100000 | 2.4719 | 2.3869   | 2.3880    |
>
> Next, we would like to address your comments one by one.
>
> **Weakness**
>
> > The number of experiments is limited...
>
> We acknowledge that the original submission primarily focused on synthetic experiments, which limited the scope of practical evaluation. To address this concern, we conducted additional experiments after submission, including both real-world market data and more complex hedging scenarios. These new results, as introduced in our text above, provide a broader evaluation of our method and could address the concerns in the following points.
>
> >It is unclear how well the OOD experiment in Figure 1 reflects real-world financial data...
>
> The OOD setting in Figure 1 was designed to highlight the robustness properties of the proposed strategy under distributional shifts, though it may not fully capture the complexity of real financial markets. To address this limitation, we conducted an additional experiment using historical market data as introduced in **Experiment 1**. This naturally introduces realistic and substantial distribution shifts. The results confirm that our method remains effective under real-world market uncertainty, reinforcing its practical relevance.
>
> >...when the number of training samples is sufficiently large, there is little to no performance gap between robust training and clean training...
>
> Such an observation is reasonable and consistent with our findings when the underlying model is relatively simple. In these cases, we observe that the performance of clean strategies plateaus as the sample size increases, indicating that the empirical distribution with 100,000 samples closely approximates the true distribution, thereby reducing the observed benefit of robust training. In the second experiment introduced above, we incorporated transaction costs into the hedging problem, adding nonlinearity and increasing the overall complexity. Under this more realistic setting, we observe that performance no longer plateaus as the sample size increases, and adversarial training continues to outperform clean training even with 100,000 samples. This suggests that the advantage of robust training is not limited to low-data regimes but also extends to more complex or imperfectly specified environments, where model uncertainty remains significant despite the availability of large training datasets.
>
> **Question**
>
> >...whether real-world market uncertainty—such as the COVID-19 crisis or post-2008 volatility—could be used as a more realistic evaluation. For example, training on pre-crisis data and testing on crisis-period data...
>
> The additional **Experiment 1** directly addresses this question by training on pre-COVID historical data and testing on real market data from March 2020. In this case, adversarially trained strategies continue to outperform standard training.

---

> ### Author Response · Authors · 2025-08-07
>
> Thank you again for your review. We wanted to kindly follow up on the rebuttal we provided. If possible, we would greatly appreciate any feedback you may have at this stage. If there are any remaining questions or points you’d like to discuss, we’d be very happy to provide further clarification.
>
> We sincerely appreciate your time and feedback.

---

> > ### Comment · Reviewer_P7ou · 2025-08-08
> > **Thank your rebuttal.**
> >
> > Thank you for your response.
> > I think the consideration of adversarial robustness in the context of financial engineering particularly interesting.Therefore, I will maintain my score. (border accept)

---

> > > ### Author Response · Authors · 2025-08-09
> > >
> > > Thank you for your response. We appreciate your thoughtful review and for maintaining a positive score.

---

### Official Review · Reviewer_RmzV · 2025-07-03

**Clarity:** 2
**Significance:** 2
**Originality:** 2
**Rating:** 4
**Confidence:** 4

**Summary:**

This paper investigates the robustness of deep hedging strategies, a data-driven approach for hedging financial derivatives. The authors first demonstrate that standard deep hedging models are highly susceptible to small adversarial perturbations in the underlying data distribution, which can lead to significant performance degradation. To address this vulnerability, the paper proposes a novel adversarial training framework specifically designed for deep hedging.
The core of their contribution is framing the problem within Distributionally Robust Optimization (DRO). They formulate a min-max optimization problem where a hedging agent minimizes a risk measure while an adversary searches for the worst-case distribution within a Wasserstein ball around the nominal data distribution. Since this problem is computationally intractable, the authors derive a tractable approximation by reformulating the search for the worst-case distribution as a direct perturbation of the empirical data samples. Based on this reformulation, they develop two practical algorithms, Wasserstein Projection Gradient Descent (WPGD) and Wasserstein Budget Projection Gradient Descent (WBPGD), to generate these adversarial distributions.
Through extensive experiments on the Heston model, the authors show that their adversarially trained strategies are not only more resilient to these attacks but also achieve substantially better out-of-sample and out-of-distribution performance, especially when the training dataset is small.

**Questions:**

Refer to the section on weaknesses. I seek an interactive dialogue with authors during the rebuttal phase. If the concerns outlined in the weaknesses section are addressed thoroughly, I am willing to consider raising my score.

**Ethical Concerns:**

["NO or VERY MINOR ethics concerns only"]

**Final Justification:**

The exploration of adversarial robustness within the realm of financial engineering is particularly noteworthy. Consequently, I shall strive to enhance my score.  (border acceptance)

However, I do not consider the paper to be strong enough to be accepted to the NIPS conference.

**Limitations:**

Yes

**Paper Formatting Concerns:**

No formatting issues are evident.

**Quality:**

2

**Strengths And Weaknesses:**

**Strengths:**

The paper addresses a critical and practical problem: the robustness of deep hedging strategies to model misspecification. While deep hedging has gained popularity, its performance relies heavily on the quality of the training data. This work is the first to systematically unify distributional adversarial training with the deep hedging framework, providing a principled method to enhance robustness against uncertainties in financial markets. The authors ground their approach in the well-established theory of Distributionally Robust Optimization (DRO). The key theoretical contribution is the computationally tractable reformulation of the adversarial optimization problem over a Wasserstein ball. By leveraging sensitivity analysis results and connecting them to the deep hedging problem, they provide a practical and theoretically justified algorithm for a problem that is otherwise intractable due to the non-convexity of the neural network loss function.

The experimental validation is comprehensive and well-designed. The authors first clearly establish the problem by showing that standard deep hedging strategies are indeed vulnerable to their proposed attacks, with performance degrading significantly under minor, statistically plausible perturbations. The results on adversarial training are particularly strong. The finding that robust training significantly improves out-of-sample and out-of-distribution performance, especially in low-data regimes (e.g., a 54% lower mean hedging loss with N=5,000 samples), is a powerful demonstration of the method's practical benefit. This suggests the method acts as an effective regularizer.
The evaluation on an out-of-distribution dataset generated with perturbed model parameters further strengthens the claim of generalizability and robustness.

**Weaknesses:**

While the paper is strong, there are a few areas where it could be improved or that merit further discussion. The core of the proposed method relies on approximating the infinite-dimensional Wasserstein DRO problem with a finite-dimensional optimization over the empirical samples (Equation 3.14). This approximation is shown to be accurate for an infinitesimally small perturbation radius
 δ (as δ → 0). However, the experiments use finite, non-trivial values for
 δ (up to 0.5). A discussion on the tightness of this approximation for the radii used in the experiments would strengthen the paper's claims. While the empirical results are strong, it is unclear how much performance is left on the table compared to solving the true (but intractable) DRO problem. The authors rightly identify the sensitivity to the choice of the Wasserstein radius δ as a limitation. This is a crucial practical point. The paper presents results for adversarially trained models with
 δ=0.01 and δ=0.1 but does not detail how these values were chosen. A sensitivity analysis or discussion on a principled method for selecting
 δ (e.g., via a validation set) would significantly increase the practical value of this work. The results in Table 3 clearly show the trade-off between nominal performance and robustness, and more guidance on navigating this trade-off would be beneficial. The paper is motivated by the general problem of "model misspecification" and "distributional shifts". The proposed adversarial training provides robustness against a specific, worst-case perturbation within a Wasserstein ball. While powerful, this may not cover all types of realistic distribution shifts in financial markets, such as sudden regime changes or structural breaks that might alter the underlying data-generating process in ways not well-captured by a Wasserstein neighborhood. Acknowledging this limitation and contextualizing the specific nature of the robustness achieved would be helpful.

---

> ### Author Rebuttal · Authors · 2025-07-30
>
> Dear reviewer,
>
> Thank you for your time and effort in reviewing our paper and for your fruitful suggestions.
>
> To start with, we would like to share 2 new experiments conducted after our submission.
>
> **Experiment 1: GAD process with real-market data**
>
> In the paper [11], the authors investigate hedging strategies trained on synthetic data generated from a generalized affine diffusion (GAD) process and evaluated on real market prices of leading S&P 500 companies during the COVID-19 crash in March 2020. Following their setup, we construct two training datasets: FIX, where model parameters are fixed based on historical pre-COVID estimates, and ROB, where parameters are sampled from intervals inferred from the same historical data to incorporate model uncertainty. Clean and adversarial training are applied to both datasets, using the entropic risk measure the same as in the Black-Scholes model. As there is no validation set to choose hyperparameters, for adversarial training, we fix the sample size to $N=100,000$ and take the average across six $(α, δ)$ pairs. We provide the average P\&L(eq2.2) below.
>
> | Method| Dataset | AAPL| AMZN| BRK-B| GOOGL| MSFT|
> |-|-|-|-|-|-|-|
> | Clean| FIX| -2.261  | -0.981  | -2.086  | -1.275  | -4.341  |
> | Clean| ROB| -0.830  | -0.849  | -0.281  | 0.026   | -0.223  |
> | Adv  | FIX| **-0.579**  | **-0.291**  | **-0.127**  | -0.459  | -0.564  |
> | Adv  | ROB| -0.739  | -0.860  | -0.294  | **0.199**  | **-0.144**  |
>
> The results show that clean training on the ROB dataset outperforms FIX, confirming the findings of [11]. More importantly, adversarial training consistently improves performance across all settings. Even when ROB-clean outperforms FIX-clean, adversarial training on ROB yields further gains. This highlights the strength of adversarial training in improving robustness under model misspecification.
>
> This setup captures not only the distributional shift caused by market volatility but also the mismatch between assumed model dynamics and real-world behavior, offering a meaningful test of robustness. Our results demonstrate that adversarially trained strategies outperform standard training in this challenging real-market scenario, supporting the practical value of adversarial training methods we proposed.
>
> **Experiment 2: Heston model with transaction cost**
>
> In addition, we extend our study to the Heston model with transaction costs by incorporating an additional cost term into the loss function. The experimental setup otherwise remains identical to the cost-free case. As shown in the following table, like in Fig.1, the adversarially trained strategy consistently outperforms the clean one across all sample sizes. Interestingly, unlike the case without transaction costs—where performance plateaus beyond a certain data size—the hedging loss here continues to decline as the number of training samples increases. This suggests that even with 100,000 samples, the data may not fully represent the true distribution. In such scenarios, adversarial training provides a clear advantage, underscoring its potential to improve robustness under data scarcity or distributional uncertainty.
>
> | N      | clean  | S-attack | SV-attack |
> |-|-|-|-|
> | 5000   | 2.8441 | 2.8237   | 2.8249    |
> | 10000  | 2.7993 | 2.7919   | 2.7929    |
> | 20000  | 2.7606 | 2.7510   | 2.7511    |
> | 50000  | 2.6419 | 2.6064   | 2.6137    |
> | 100000 | 2.4719 | 2.3869   | 2.3880    |
>
> Next, we would like to address your comments one by one.
>
> **Weaknesses**
> >... However, the experiments use finite, non-trivial values for δ (up to 0.5). A discussion on the tightness of this approximation for the radii used in the experiments would strengthen the paper's claims. While the empirical results are strong, it is unclear how much performance is left on the table compared to solving the true (but intractable) DRO problem...
>
> As discussed in Equation 3.13, the proposed algorithm is a tractable approximation of the infinite-dimensional Wasserstein DRO problem and recovers the true objective up to
> o(δ) as δ→0. This theoretical guarantee provides a first-order justification for our formulation.
>
> We acknowledge, however, that in practice we use finite (and sometimes moderately large) values of δ, and the approximation error for such values is not explicitly characterized. Although we do not provide a formal characterization of the approximation error at finite radii, the empirical results suggest that the method remains effective. Quantifying the tightness of the approximation for finite δ is indeed an important question, and we agree that further theoretical or empirical analysis of this aspect could strengthen the robustness guarantees of the method. We consider this a promising direction for future research.
>
> >The paper presents results for adversarially trained models with δ=0.01 and δ=0.1 but does not detail how these values were chosen. A sensitivity analysis or discussion on a principled method for selecting δ (e.g., via a validation set) would significantly increase the practical value of this work. The results in Table 3 clearly show the trade-off between nominal performance and robustness, and more guidance on navigating this trade-off would be beneficial.
>
> We would first like to clarify that our approach to hyperparameter selection, including the choice of the Wasserstein radius δ, is described in Appendix D. For experiments on simulated data, we perform a grid search over a range of values and select the configuration that performs best on a held-out validation set.
>
> Regarding Table 3, we believe there may have been a misunderstanding. This table is not intended to evaluate the out-of-sample performance of the trained strategies with optimized hyperparameters, but rather to illustrate how different strategies respond to adversarial attacks of increasing strength. The losses shown are evaluated on the training data, and the columns represent the magnitude of the adversarial perturbation applied at test time (not the training δ). As expected, the loss increases with larger perturbations across all models.
>
> In this table, we compare three strategies: clean training, adversarial training with δ = 0.01, and adversarial training with δ = 0.1. These values are not the result of hyperparameter optimization but are representative runs chosen to illustrate the trade-off between nominal performance and robustness, as discussed in Section 5.3.
>
> >... While powerful, this may not cover all types of realistic distribution shifts in financial markets, such as sudden regime changes or structural breaks that might alter the underlying data-generating process in ways not well-captured by a Wasserstein neighborhood.
>
> While our primary experiments are based on simulated data using well-established models such as Black-Scholes and Heston, we agree that evaluation on real-world data is essential for assessing practical relevance. To this end, we have included an additional experiment (see **Experiment 1** from above) using historical market data, as introduced earlier. This setting captures both the distributional shift induced by market volatility and the discrepancy between the assumed model dynamics and real-world behavior, offering a meaningful test of robustness. We believe this experiment complements the synthetic benchmarks and reinforces the method’s applicability in real financial environments.
>
> We acknowledge that robustness within a Wasserstein ball addresses a specific form of local distributional shift and may not fully capture all types of structural changes or regime shifts observed in financial markets. As discussed in the last paragraph of the main text, exploring alternative ambiguity set geometries or hybrid formulations that better account for global distributional changes is indeed a promising direction for future research.
>
> **Question**
>
> >Refer to the section on weaknesses. I seek an interactive dialogue with authors during the rebuttal phase.
>
> We sincerely appreciate the reviewer’s openness to discussion and willingness to reconsider the score. We have carefully addressed the concerns outlined in the weaknesses section in our detailed responses and welcome any further questions or clarifications during the rebuttal phase.

---

### Comment · Area_Chair_wpT4 · 2025-08-07

Dear Reviewers,

Thank you for your efforts in reviewing the submission.

The authors have submitted their rebuttals in response to your comments. Please ensure that you engage with the rebuttal and provide a response before selecting “Mandatory Acknowledgement.”

We have noticed that some reviewers have submitted a “Mandatory Acknowledgement” without posting any discussion or feedback to the authors. Please note that this is not permitted.

Please note “Mandatory Acknowledgement” button is to be submitted only when reviewers fulfill all conditions below (conditions in the acknowledgment form):
1. read the author rebuttal
2. engage in discussions (reviewers must talk to authors, and optionally to other reviewers and AC - ask questions, listen to answers, and respond to authors)
3. fill in "Final Justification" text box and update “Rating” accordingly (this can be done upon convergence - reviewer must communicate with authors first)

---

### Note · Authors · 2025-08-12

We sincerely thank the reviewers and the Area Chair for their time and valuable feedback. We have carefully addressed each comment point by point in our rebuttal.

A recurring concern in the initial reviews was that the experimental evaluation in the original submission was not sufficiently comprehensive. We share this view and, accordingly, conducted two additional experiments after submission. As detailed in our response, one experiment uses real market data, incorporating market shifts during the COVID period as highlighted by some reviewers. The other evaluates the Heston model with transaction costs, adding further realism and robustness to the evaluation. These new results complement the original experiments, strengthen the empirical evidence, and broaden the applicability of our proposed method.

We also appreciate the constructive suggestion from one reviewer regarding the paper’s structure, which will be incorporated into the revised version to improve clarity and readability.

Several reviewers have noted the relevance and significance of our work. We believe that the additional experiments, clarifications, and planned improvements have substantially strengthened the paper and addressed the reviewers’ comments. We hope that the revised work will be seen as a clearer and more compelling contribution.

---

### Decision · Program_Chairs · 2025-09-17

**Decision:**

Accept (poster)

**Comment:**

This paper introduces a distributional adversarial training framework for deep hedging, grounded in Wasserstein DRO. The method is novel, theoretically motivated, and empirically effective, with added experiments on real market data and transaction costs. While some concerns on scope and clarity were raised, the rebuttal addressed them satisfactorily. Overall, this is a solid and timely contribution.